# Spatial Seismic Hazard Variation and Adaptive Sampling of Portfolio Location Uncertainty in Probabilistic Seismic Risk Analysis

Christoph Scheingraber[1] and Martin Käser[2,1]

[1]Ludwig-Maximilians-Universität, Munich, Germany
[2]Munich Reinsurance, Munich, Germany

**Correspondence:** Christoph Scheingraber (scheingraber@geophysik.uni-muenchen.de)

**Abstract.** Probabilistic Seismic Risk Analysis is widely used in the insurance industry to model the likelihood and severity of losses to insured portfolios by earthquake events. The available ground motion data - especially for strong and infrequent earthquakes - are often limited to a few decades, resulting in incomplete earthquake catalogues and related uncertainties and assumptions. The situation is further aggravated by the sometimes poor data quality with regard to insured portfolios. For example, due to geocoding issues of address information, risk items are often only known to be located within an administrative geographical zone, but precise coordinates remain unknown to the modeler.

We analyze spatial seismic hazard and loss rate variation inside administrative geographical zones in western Indonesia. We find that the variation of hazard can vary strongly between different zones. The spatial variation of loss rate displays a similar pattern as the variation of hazard, without depending on the return period.

In a recent work, we introduced a framework for stochastic treatment of portfolio location uncertainty. This results in the necessity to simulate ground motion on a high number of sampled geographical coordinates, which typically dominates the computational effort in Probabilistic Seismic Risk Analysis. We therefore propose a novel sampling scheme to improve the efficiency of stochastic portfolio location uncertainty treatment. Depending on risk item properties and measures of spatial loss rate variation, the scheme dynamically adapts the location sample size individually for insured risk items. We analyze the convergence and variance reduction of the scheme empirically. The results show that the scheme can improve the efficiency of the estimation of loss frequency curves, and may thereby help to spread the treatment and communication of uncertainty in Probabilistic Seismic Risk Analysis.

## 1 Introduction

Seismic risk analysis is widely used in academia and industry to model the possible consequences of future earthquake events, but is often limited by the availability of reliable earthquake event ground motion data over a longer period of time, resulting in the necessity for many assumptions and a wide range of deep uncertainties (Goda and Ren, 2010). The treatment and communication of uncertainties is highly important for informed decision making and a holistic view of risk (Tesfamariam et al., 2010; Cox, 2012; Bier and Lin, 2013). In the insurance industry, Probabilistic Seismic Risk Analysis (PSRA) is the

means of choice to model the likelihood and severity of losses to insured portfolios due to earthquakes. In this context precise exposure locations are often unknown, which can have a significant impact on scenario loss, as well as on loss frequency curves (Bal et al., 2010; Scheingraber and Käser, 2019).

For PSRA in the insurance industry, uncertainty is usually taken into account by means of Monte Carlo (MC) simulation (e.g. Pagani et al. 2014; Tyagunov et al. 2014; Foulser-Piggott et al. 2017). This is a computationally intensive process, because the error convergence of MC is relatively slow and a high-dimensional loss integral needs to be evaluated with a sufficient sample size. In PSRA, the hazard component typically dominates the overall model runtime. As a result, stochastic treatment of portfolio location uncertainty can be particularly challenging - ground motion needs to be simulated on a large number of sampled risk locations. On the other hand, a fast model runtime is a key requirement for underwriting purposes in the insurance industry. Methods or sampling schemes to improve the error convergence of MC simulation are known as variance reduction techniques. MC simulation is ubiquitous in many areas of science and engineering and a wide variety of sampling schemes exists. Some well-known ideas are common random numbers and control variates (Yang and Nelson, 1991), importance-, stratified- and hypercube sampling, Quasi Monte Carlo Simulation (QMC) using low-discrepancy sequences, as well as adaptive sampling. The error convergence of different sampling schemes has been investigated for many different types of integrals and application areas (Hess et al., 2006; dos Santos and Beck, 2015). Some work has already been performed on variance reduction for PSHA and PSRA in the form of importance sampling, e.g. preferentially sampling the tails of the magnitude and site ground motion probability distributions (Jayaram and Baker, 2010; Eads et al., 2013). However, to our knowledge so far no study has specifically investigated variance reduction for location uncertainty in PSRA in a modern risk assessment framework. Building on a framework proposed in a recent study, in the present paper we describe a novel variance reduction scheme specifically designed to increase the computational efficiency of stochastic treatment of portfolio location uncertainty in PSRA.

The remainder of this paper is structured as follows. We outline the most important theoretical background in Section 2. Using a seismic risk model of western Indonesia, in Section 3 we explore spatial hazard and loss rate variation inside administrative zones. Based on this, in Section 4 we propose an adaptive location uncertainty sampling scheme and investigate its performance using several test cases in Section 5. In Section 6, we give some recommendations on how to apply the results in practice and conclude with possible future improvements.

## 2 Background

### 2.1 Seismic Hazard and Risk Analysis

Seismic hazard is modeled in both academy and industry using a variety of different methods. Deterministic Seismic Hazard Assessment traditionally aims at identifying the "maximum creditable earthquake" that represents the largest seismic hazard to a particular site. This approach is often used to estimate the so-called "design earthquake ground motion" for earthquake engineering purposes (e.g. Mualchin 2011; Tsapanos et al. 2011). Another approach is to employ extreme value statistics to assess the probability of rare events, e.g. earthquakes that are more severe than those historically observed (e.g. Papadopoulou-

Vrynioti et al. 2013; Pavlou et al. 2013). In this study we use PSRA. Below, we briefly outline the underlying theory and give reasons why this approach is well-suited for the purposes of the insurance industry.

PSRA is based on the method of Probabilistic Seismic Hazard Analysis (PSHA; Cornell, 1968; Senior Seismic Hazard Committee, 1997), which relies on a number of assumptions outlined in the following. In PSHA, the exceedance rate $\lambda$ of ground motion level $y_0$ at a site $\boldsymbol{r}_0$ is expressed by the hazard integral

$$\lambda(y_0, \boldsymbol{r}_0)[y \geq y_0] = \int\limits_{V} \int\limits_{m_{\min}}^{m_{\max}} P[y \geq y_0|m, \boldsymbol{r}, \boldsymbol{r}_0] \cdot \nu(m, \boldsymbol{r}) dm d\boldsymbol{r}, \tag{1}$$

with $\nu(m, \boldsymbol{r}) dm d\boldsymbol{r}$ the seismic rate density which describes the spatio-temporal distribution of seismic activity, $P[y \geq y_0|m, \boldsymbol{r}, \boldsymbol{r}_0]$ the conditional probability of exceeding ground motion $y_0$ at site $\boldsymbol{r}_0$ given a rupture of magnitude $m$ at source location $\boldsymbol{r}$, and $V$ the spatial integration volume containing all sources which can cause relevant ground motion at $\boldsymbol{r}_0$. Assuming that the occurrence of earthquake events is a temporal Poisson process, the probability of at least one exceedance of $y_0$ within time interval $t_0$ is given by

$$P(y_0, t_0, \bar{\lambda})[y \geq y_0] = 1 - e^{-\bar{\lambda} t_0}, \tag{2}$$

where $\bar{\lambda}$ is the mean annual recurrence rate. Classical PSHA relies on quadrature integration, i.e. a deterministic algorithm is used to solve Equation 1. A historically important software implementation that is still popular and useful for educational purposes is SEISRISK III (Bender and Perkins, 1987). Classical PSHA has its limits however, since the treatment of advanced topics, such as complex source geometries or spatial ground motion correlation, is often intricate and sometimes impossible. The approach is not well-suited to be extended to assess the risk of building damage and monetary loss to spatially distributed insurance portfolios.

For PSRA in the insurance industry, event-based MC simulation is nowadays commonly used, since this approach is well-suited to numerically solve high-dimensional integrals as required for large-scale risk analyses of insured portfolios with many uncertainties. In this approach, stochastic simulation is performed to obtain a set of stochastic ground motion fields $\hat{\mathbf{Y}}$ and to then compute the probability that a loss level $\iota_0$ is exceeded as

$$P(\hat{\mathbf{Y}}, \Theta)[\iota \geq \iota_0] = \sum_{i=1}^{n_{\mathrm{e}}} \int\limits_{\iota_0}^{\infty} f_\iota(\iota|\hat{\mathbf{Y}}_i, \Theta) d\iota, \tag{3}$$

where $f_\iota(\iota|\hat{\mathbf{Y}}_i, \Theta)$ is the loss probability density function for a portfolio $\Theta$ given the $i$th ground motion field $\hat{\mathbf{Y}}_i$. Summing up the contribution of all $n_{\mathrm{e}}$ events yields the total loss exceedance probability. A Probable Maximum Loss (PML) curve, showing loss against mean return period $T$ (with $T = 1/\bar{\lambda}$), can be obtained from the loss exceedance probability curve (Equation 3) using a first order Taylor approximation of Equation 2:

$$T = \frac{t_0}{P(y_0, t_0, \lambda)[y \geq y_0]}. \tag{4}$$

Here, $t_0$ is the period of interest (time interval), which is 1 year for most reinsurance contracts.

For more details on PSHA and PSRA, we refer to the comprehensive textbook of McGuire (2004).

## 2.2 Portfolio Location Uncertainty

Perhaps surprisingly, in the insurance industry, portfolios frequently lack precise coordinate-based location information. Obtaining this information is often not possible, e.g. because geocoding engines are not used systematically or can not reliably obtain coordinates from the policy address of the insured risk. Especially for large treaty portfolios with thousands or millions of risks, it apparently is simply too much effort for the primary insurer or the insurance broker to obtain and provide this information. Unfortunately, this is also not uncommon for smaller portfolios consisting only of a few hundred high-value risks.

However, administrative zones, such as postal codes, can easily be obtained from the insurance policy.

Exposure uncertainty has previously been identified as an important area of research (Crowley, 2014), and we already introduced a framework for stochastic treatment of location uncertainty in a recent paper (Scheingraber and Käser, 2019). In our framework, locations of risk items without precise coordinate location information are sampled with replacement from a weighted irregular grid inside their corresponding administrative zone. The grid weights are used to preferentially sample

locations in areas of assumed high insurance density, e.g. based on population density or on commercial and industrial inventory data depending on the type of risk (Dobson et al., 2000). An example of such a weighted grid is shown in Figure 1.

In MC simulation, the choice of a pseudo-random number generator is of particular importance. In this study we use *MRG32K3a*, a combined multiple recursive generator which efficiently generates random number sequences with low memory requirements and excellent statistical properties (L'Ecuyer, 1999). MRG32K3a supports up to $1.8 \cdot 10^{19}$ statistically indepen-

dent *substreams*. Each substream has a *period*[1] of $7.6 \cdot 10^{22}$. These properties make MRG32K3a well-suited for a large scale parallel MC simulation of seismic risk.

## 2.3 Evaluation of the Proposed Sampling Scheme

### 2.3.1 Standard Error

Given that MC simulation is a stochastic method, there are no strict error bounds for statistics of interest obtained from a

sample of finite size $n$. The error is therefore usually estimated using the standard deviation of the sampling distribution of the respective statistic, which is referred to as its standard error ($E_{SE}$). If the sampling distribution is known (e.g. normal), standard errors can often be obtained using a simple closed-form expression (Harding et al., 2014). For the statistics estimated in this study, e.g. PML at a specific return period, we can however not make a valid distribution assumption when taking location uncertainty into account. We therefore use repeated simulation to evaluate the performance of the proposed sampling scheme.

---

[1]The period of a pseudo-random number generator refers to the minimum length of a generated sequence before the same random numbers are repeated cyclically.

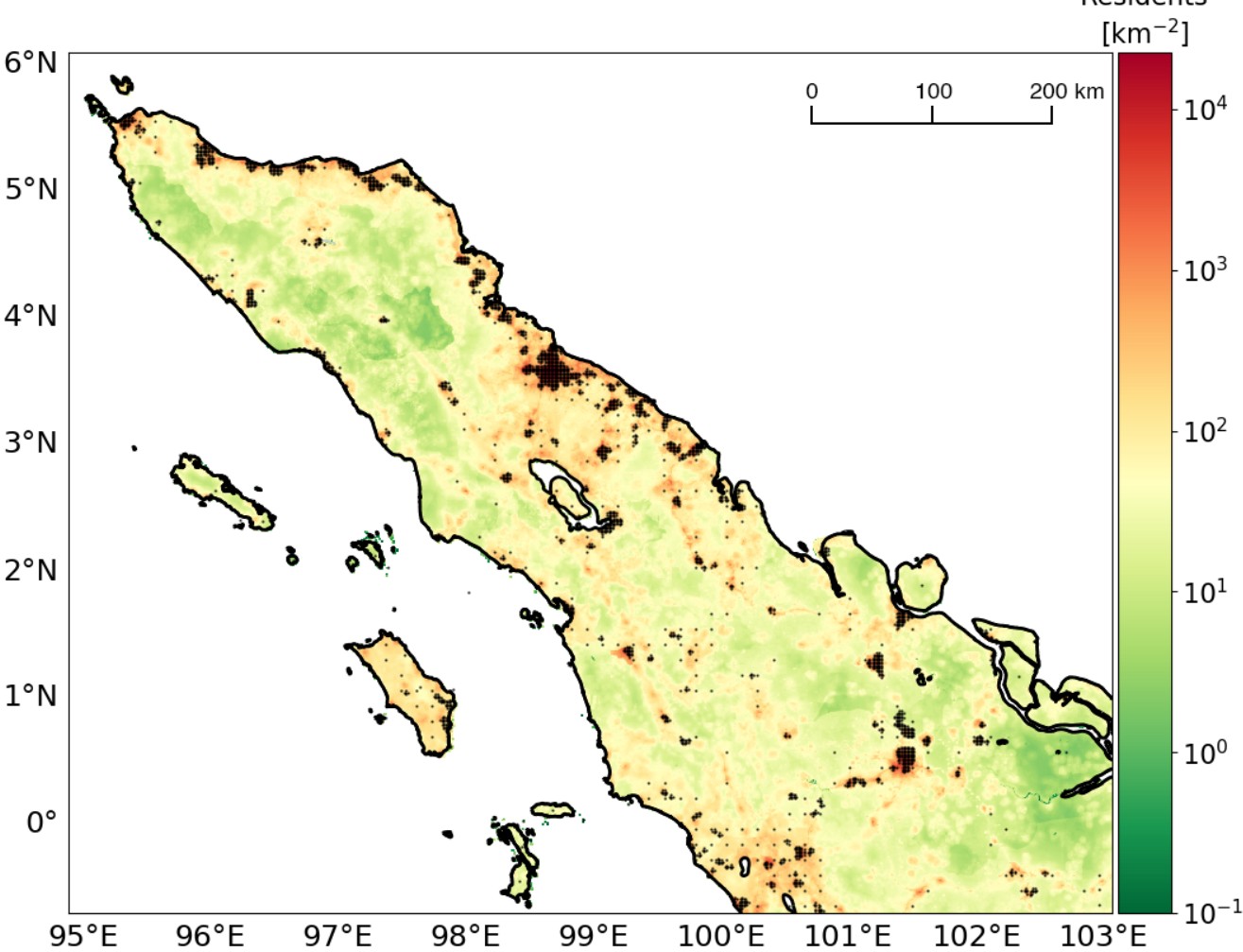

**Figure 1.** An example of a weighted grid used as an insurance density proxy for the location uncertainty framework. The map shows northern Sumatra. Color indicates population density (residents per $\mathrm{km^2}$) as a proxy for insured exposure density. Black markers depict grid points of the weighted grid. The population data in this plot is based on a free dataset (Gaughan et al., 2015).

The standard error can then be estimated as

$$\mathrm{E_{SE}}(\hat{\Phi}_R) = \sqrt{\mathrm{Var}(\hat{\Phi}_R)}, \tag{5}$$

where $\hat{\Phi}_R$ denotes a set of estimations of a statistic obtained from $R$ repeated simulations and $\mathrm{Var}(\cdot)$ the variance operator. The corresponding relative standard error $E_{\mathrm{RSE}}$ can be obtained by dividing $E_{\mathrm{SE}}$ by the estimated statistic. To estimate confidence intervals of standard errors, we use bootstrapping with the bias-corrected accelerated percentile method (Efron, 1979; Efron and Tibshirani, 1986).

### 2.3.2 Bias and Convergence Plots

The bias of an estimator $\hat{\theta}$ is defined as

$$\text{Bias}(\hat{\theta}_n) = \mathbb{E}_\theta(\hat{\theta}_n) - \theta, \tag{6}$$

where $\hat{\theta}_n = f(x_1, x_2, \ldots, x_n)$ is the estimator depending on the $n$ members of the sample and $\mathbb{E}_\theta$ its expected value. Taking into account that deriving the bias analytically is infeasible for a complex numerical simulation such as the one performed by our framework, we use simple MC[2] with a large sample size as empirical reference and approximation for $\theta$. In addition, we use convergence plots, which are a simple yet powerful method to monitor and verify the results (Robert and Casella, 2004). The values estimated using simple MC and the adaptive variance reduction scheme are plotted against increasing sample size $n$.

### 2.3.3 Variance Reduction, Convergence Order and Speedup

To quantify the performance of the proposed scheme at a particular sample size $n$, we use the following well-known definition of variance reduction VR:

$$\text{VR} = \frac{\sigma_{\text{MC}}^2}{\sigma_{\text{LSS}}^2}, \tag{7}$$

where $\sigma_{\text{MC}}^2$ is the variance using simple MC and $\sigma_{\text{LSS}}^2$ the variance using the proposed location sampling scheme (MacKay, 2005; Juneja and Kalra, 2009).

To describe the asymptotic error behavior for growing $n$, we use the big O notation ($\mathcal{O}$; Landau, 1909; Knuth, 1976). For example, the error convergence order of simple MC is always $\mathcal{O}(n^{-0.5})$, independent of the dimensionality of the integrand (Papageorgiou, 2003).

To compare the real runtime required by simple MC and the proposed scheme to reach a specific relative standard error level $\varepsilon_{\text{RSE}}$, we use the speedup S defined as

$$\text{S} = \frac{t_{\text{MC}}}{t_{\text{LSS}}}, \tag{8}$$

where $t_{\text{MC}}$ the runtime required by simple MC and $t_{\text{LSS}}$ the runtime required by the proposed location sampling scheme.

## 2.4 Generation of Synthetic Portfolios

In this work, we use synthetic portfolios in western Indonesia modeled after real-world counterparts in terms of spatial distribution of risk items, as well as value distribution among risk items.

### 2.4.1 Value Distribution

The total sum insured (TSI) is kept constant for all portfolios:

$$\text{TSI} = \text{const.} = 1 \cdot 10^6. \tag{9}$$

---

[2]For simple MC, the strong law of large numbers guarantees an *almost certain* convergence for $n \to \infty$.

However, the TSI is distributed among a varying number of risk items (portfolio size). For this study, we use portfolio sizes $n_r$ of 1, 10, 20, 50, 100, 1000 and 10000 risk items.

The value distribution observed in many real residential portfolios can be approximated well by a randomly perturbed flat value distribution:

$$VI^*_{flat,i} = \frac{TSI}{n_r} \cdot X_i, \tag{10}$$

$$VI_{flat,i} = \frac{TSI}{\sum_{i=1}^{n_r} VI^*_{flat,i}} \cdot VI^*_{flat,i}. \tag{11}$$

$VI_{flat,i}$ ("value insured") is the value assigned to the $i$th risk item and $n_r$ denotes the number of risk items. $X_i$ is a uniform random number in the interval $[1-p, 1+p]$, where $p$ is a perturbation factor controlling the variation of insured values among individual risk items in the portfolio. For this study, we set $p$ to $0.2$. This value corresponds to the characteristics we observe in many real residential portfolios and is in accordance with the assumption of a relatively flat value distribution, which is commonly made when modeling the value distribution of residential building stock (e.g. Kleist et al. 2006; Okada et al. 2011).

Equation 11 normalizes the $n_r$ randomly perturbed insured values to ensure $\sum_{i=1}^{n_r} VI_{flat,i} = TSI$.

### 2.4.2 Geographical Distribution

For each portfolio size, we created a set of 6 portfolios with an increasing fraction of unknown coordinates: 0%, 20%, 40%, 60%, 80% and 100% of the risk items have unknown coordinates and are only known on the basis of their administrative zone (Indonesian provinces, or regencies and cities, see Section 3).

The geographical distribution of the exposure locations follows the weighted irregular grid described in Section 2.2. For each portfolio size, a portfolio with 0% unknown coordinates is initially created by choosing exposure locations from the irregular grid according to the grid point weights. For the other portfolios with the same number of risk items but a higher fraction of unknown coordinates, coordinate-based location information is then removed stepwise from the initial portfolio. In each step, 20% of the risk items are randomly selected for the removal of coordinates until all risk items have unknown coordinates.

## 3 Case Study: Spatial Seismic Hazard Variation in Western Indonesia

### 3.1 Hazard Model

For this study, we use a custom seismic risk model. Our model is based on the South-East Asia hazard model by Petersen et al. (2007) of the United States Geological Survey (USGS), which was the most recent, reliable and publicly available model when we created our risk model. Site conditions or soil classes are based on topographic slope (Wald and Allen, 2007), but have been refined locally to consider areas of soft soil, such as river beds. The geometry of the Sumatra subduction zone has been improved over the original USGS model. It is a complex representation of the fault geometry based on the *Slab 1.0* model (Hayes et al., 2012), which provides three-dimensional data of the subduction. To generate finite geometry patches for individual events on the complex fault, we use a rupture floating mechanism similar to the implementation of OpenQuake

**Table 1.** Ground motion models and corresponding weights used by the hazard model for different tectonic region types.

| Tectonic Region Type | Ground Motion Model | Weight |
|---|---|---|
| *Active Shallow Crust* | Boore and Atkinson (2008) | 1/3 |
| | Campbell and Bozorgnia (2008) | 1/3 |
| | Chiou and Youngs (2008) | 1/3 |
| *Stable Continental Crust* | Toro et al. (1997) | 1 |
| *Subduction Interface* | Zhao et al. (2006) | 2/3 |
| | Youngs et al. (1997) | 1/3 |
| *Subduction Intraslab* | Atkinson and Boore (2003) | 1 |

(Pagani et al., 2014), a free and open-source seismic hazard and risk software developed as part of the Global Earthquake Model initiative (Crowley et al., 2013).

In order to reduce the computational effort of the hazard model, we have simplified the logic tree of the ground motion model for all tectonic region types. This allowed to thoroughly analyze the performance of the proposed sampling scheme for a larger set of insurance portfolios and sampling scheme parameters, while retaining overall accordance with the hazard of the original model. Table 1 gives an overview of the selected ground motion models and their weights.

Figure 2 shows the resulting seismic hazard map for an exceedance probability of 10% in 50 years. Assuming a temporal Poisson distribution, this probability equals an average return period of 475 years, which is used for most seismic hazard maps and by the engineering community for the design of building codes. The depicted area is comprised of the islands of Sumatra, Java, and Kalimantan (the Indonesian sector on the island of Borneo). The hazard results of our model are in general agreement with results obtained using the original USGS South-East Asia model (Petersen et al., 2007) or third-party implementations, such as by the Global Earthquake Model (GEM) initiative evaluated using the OpenQuake engine[3]. Slight differences are caused by the inclusion of site conditions, simplified ground motion model logic tree, different source parameterization, improved subduction geometry representation and slightly modified seismicity rates based on the latest ISC-GEM Global Instrumental Earthquake Catalogue (Storchak et al., 2013) and Global Historical Earthquake Catalogue[4].

On the island of Sumatra, the effect of the Sumatra fault zone is clearly visible and seismic hazard is high. The seismic hazard of Java is highest in the western area of the island, around the city of Jakarta. The hazard levels decrease towards the east. On Kalimantan, there are no known or modeled seismic crustal faults. Therefore, only a low level of spatially homogenous distributed gridded seismicity is used in this area, resulting in low hazard.

The hazard model and results are described in more detail in a recent paper (Scheingraber and Käser, 2019).

---

[3]See https://hazardwiki.openquake.org/sea2007_intro for results obtained using OpenQuake.
[4]See http://www.emidius.eu/GEH/.

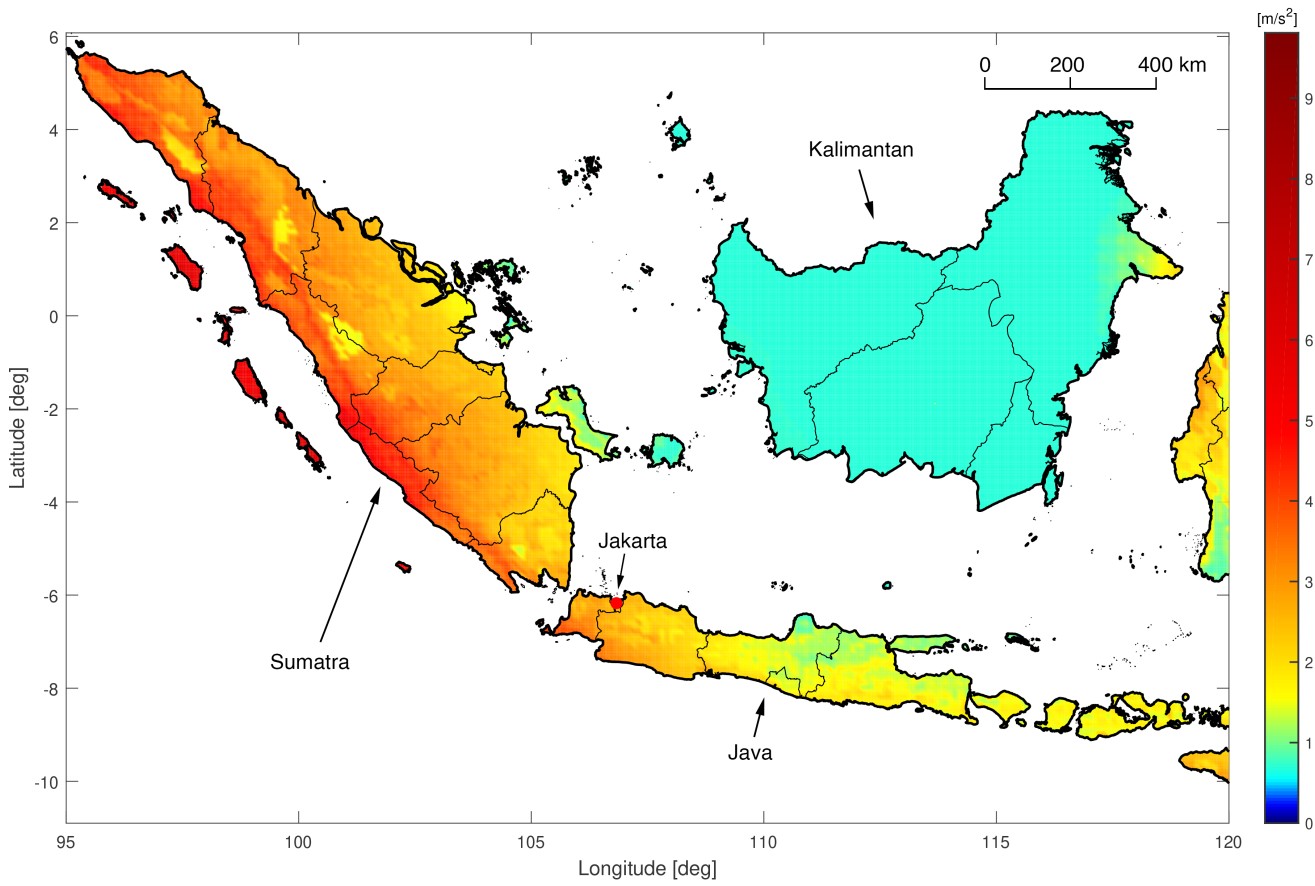

**Figure 2.** Seismic hazard in western Indonesia, as obtained using our hazard model. Color shows the level of Peak Ground Acceleration (PGA, in $\mathrm{m\,s^{-2}}$) that is predicated to be exceeded at an average return period of 475 years.

### 3.2 Spatial Seismic Hazard Variation

For this analysis, we compute seismic hazard on a regular grid using a resolution of $0.3°$. We investigate the coefficient of variation (CV) of hazard inside administrative geographical zones for different levels of resolution, corresponding to provinces and regencies or cities in Indonesia. The CV is defined as

$$\text{CV} = \frac{\sigma}{\mu}, \tag{12}$$

where $\sigma$ is the standard deviation and $\mu$ the mean.

#### 3.2.1 Dependence on Resolution Level of Geographical Zones

Figure 3 shows the CV of peak ground acceleration with an exceedance probability of 10% in 50 years per province in Indonesia. There is a noticeable decrease of the CV from west to east. The subduction modeled by the complex fault and the

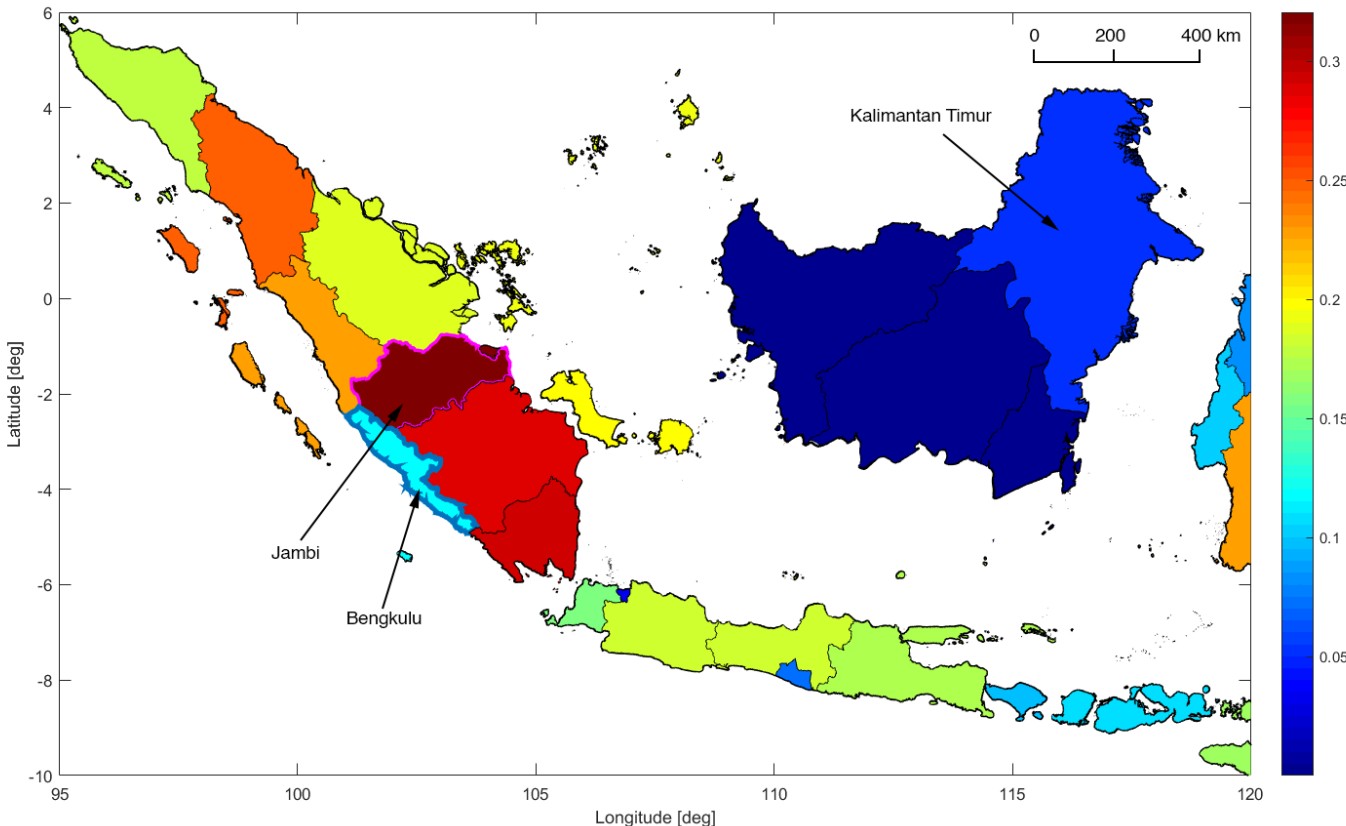

**Figure 3.** Coefficient of Variation (CV) of Peak Ground Acceleration (PGA) with an exceedance probability of 10% in 50 years inside provinces in western Indonesia. Color denotes the CV. Note how the CV is higher in provinces that have a large extent perpendicular to the Sumatra Fault Zone, such as *Jambi* (outlined in pink color), than in provinces with a small extent in that direction, such as *Bengkulu* (outlined in blue).

Sumatra Fault Zone (SFZ) result in the highest CV on Sumatra (most values 0.2 - 0.3). The CV is also relatively high on Java (around 0.15). The CV is the lowest in Kalimantan ($< 0.1$) due to the absence of any known or modeled crustal faults. As only gridded seismicity is used in this area, the hazard variation is very small. Furthermore, zones with a large extent perpendicular to the SFZ show a larger CV than zones with a smaller extent along the direction of the steepest hazard gradient. An example of this are the provinces of *Jambi* and *Bengkulu* in Figure 3. Arguably, location uncertainty is more important in *Jambi* than in *Bengkulu*.

Figure 4 shows the CV per regency or city for the same exceedance probability. Due to the smaller spatial extent of the administrative zones, the CV is in general lower at this more granular resolution of administrative geographical zones. Another observation is that the influence of individual seismo-tectonic features emerges; the CV is higher in the vicinity of modeled

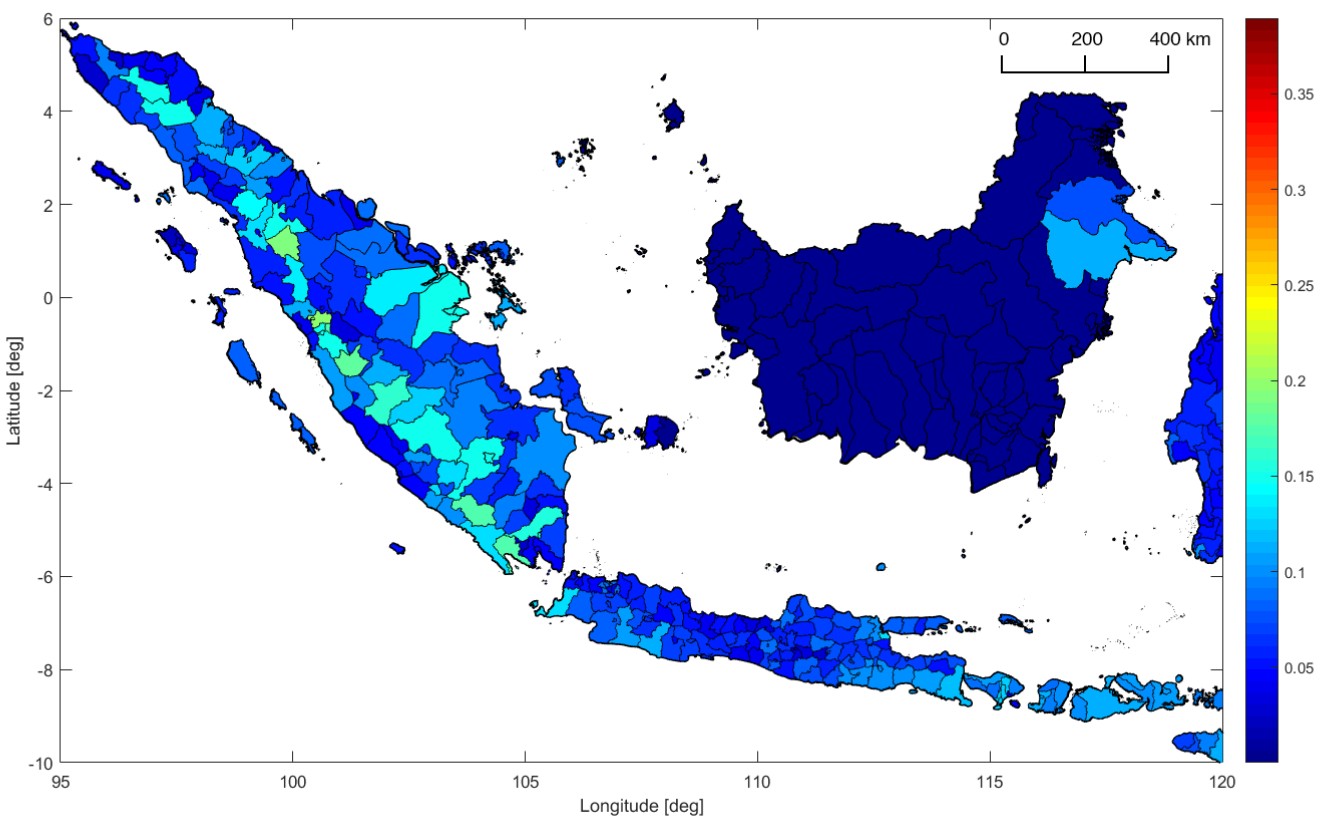

**Figure 4.** Coefficient of Variation (CV) of Peak Ground Acceleration (PGA) with an exceedance probability of 10% in 50 years inside regencies and cities in western Indonesia. Color denotes the CV. At this geographical resolution the CV is lower than for provinces (see Figure 3), and the influence of individual seismo-tectonic features, such as the Sumatra fault zone, becomes apparent.

faults. While the Sumatra subduction only has a weak influence, the SFZ has a pronounced effect. Near the SFZ, the CV has values of about $0.1$ - $0.2$. Perpendicular to the SFZ, the CV quickly drops below $0.1$.

In general, the CV is highest in zones close to modeled faults of shallow depths, as they result in a higher spatial hazard gradient compared to areas where hazard is dominated by rather regularly distributed gridded seismicity. A reasonable assumption is that location uncertainty can be particularly high in such zones.

### 3.2.2 Dependence on Return Period

Analysis of the CV across different return periods for individual zones revealed a similar pattern for most administrative zones. The CV is small for short return periods and reaches a relatively stable level above a certain return period. An example of this behavior is shown in Figure 5 for the province of *Jambi*. However, the CV does not show this pattern in all administrative

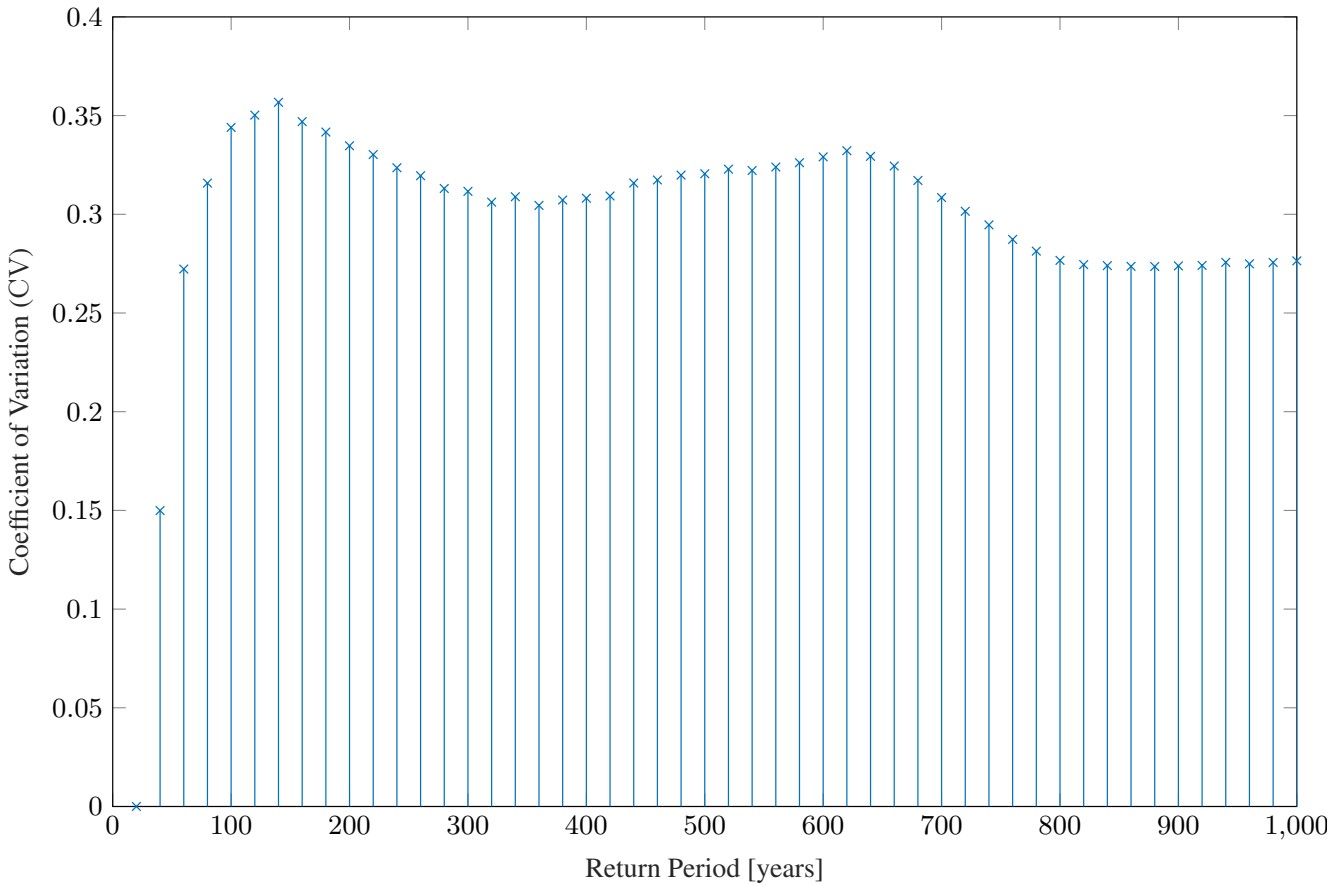

**Figure 5.** Coefficient of variation (CV) of ground motion predicted to be exceeded at various return periods for the *Jambi* province (see Figure 3). The CV remains quite stable over a large range of return periods.

zones. For some zones, especially at the level of regencies and cities, we could not determine a range of return periods for which the CV is roughly constant, as for example in the province of *Kalimantan Timur* shown in Figure 6.

### 3.3 Loss Rate Variation

The variability of the CV over return periods for certain zones makes it difficult to choose a general return period suitable for assessing the spatial variation of hazard inside a zone. To avoid the subjectivity introduced by a manual decision process for a suitable return period, we use the CV of the loss rate per zone, as it considers all return periods. Figure 7 shows the CV of the loss rate for Indonesian provinces. The overall pattern agrees with the pattern of the spatial hazard variation in Figure 3, but the range of values is much higher, from about $0.1$ to $0.9$.

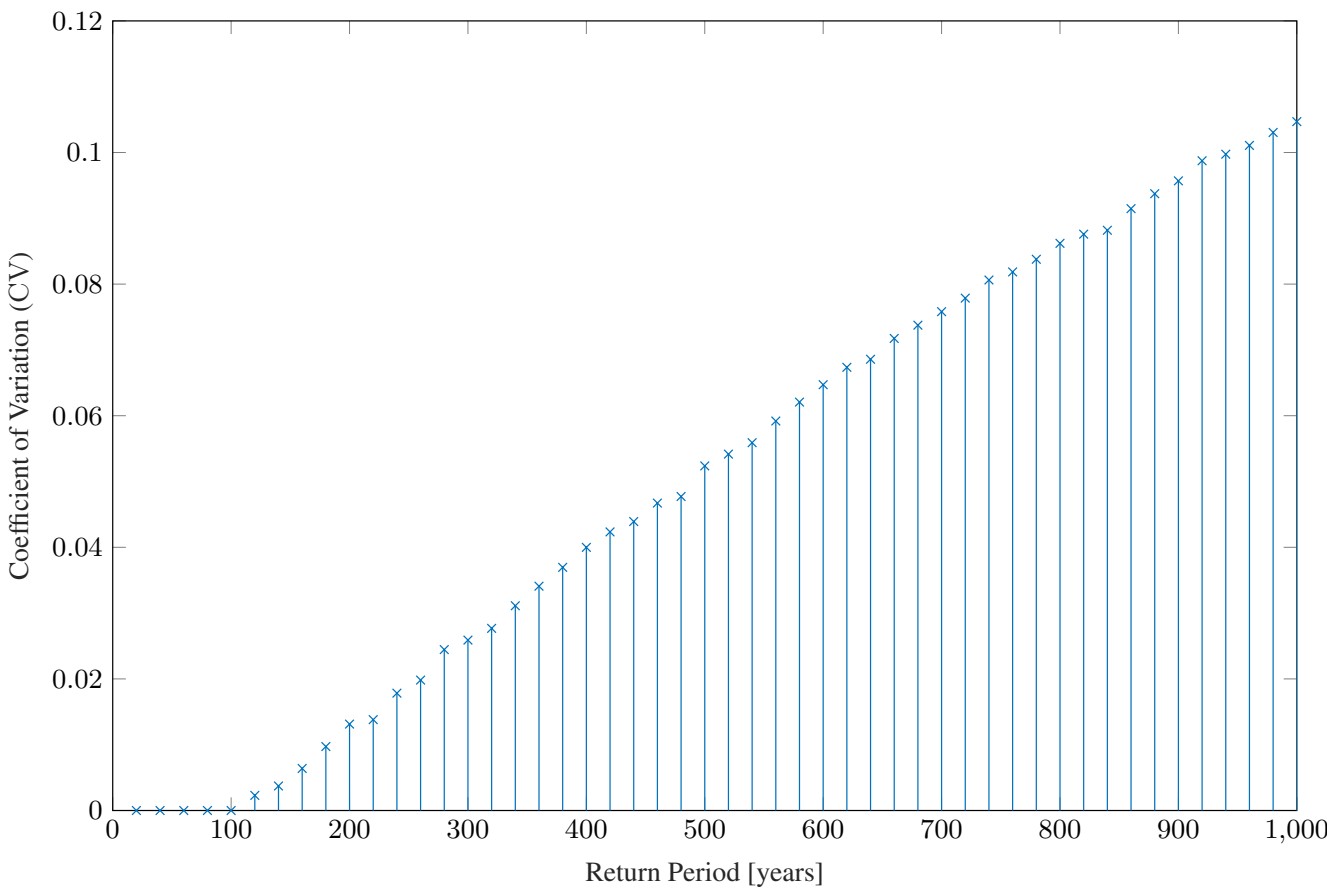

**Figure 6.** Coefficient of variation (CV) of ground motion predicted to be exceeded at various return periods for the *Kalimantan Timur* province (see Figure 3). In this case, it is not possible to determine a range of return periods for which the CV remains in a stable range.

## 4 A Framework for Adaptive Sampling of Portfolio Location Uncertainty

To increase efficiency, in our framework ground motion is jointly simulated on all unique locations of all sampled location sets. Since the computation of hazard dominates the overall runtime of PSRA, it is worthwhile to explore possibilities to distribute the number of locations on which hazard is computed in a smart way among risk items. To this end, we introduce three sampling criteria to determine the location sample size individually per risk item. A large location sample size is used for risk items for which at all three criterions indicate that location uncertainty has a strong influence. If any of the three criteria predicts that the location uncertainty has a lesser effect, a smaller sample size is used. In this way, more computational effort is invested where it is important and a better estimation of the PML curve associated with a lower variance is obtained for a given number of used hazard locations. To not add noticeable overhead to the calculation, a key requirement is that all criteria can be evaluated very efficiently. To keep the computational overhead small, another design goal is that the framework is adaptive in a sense

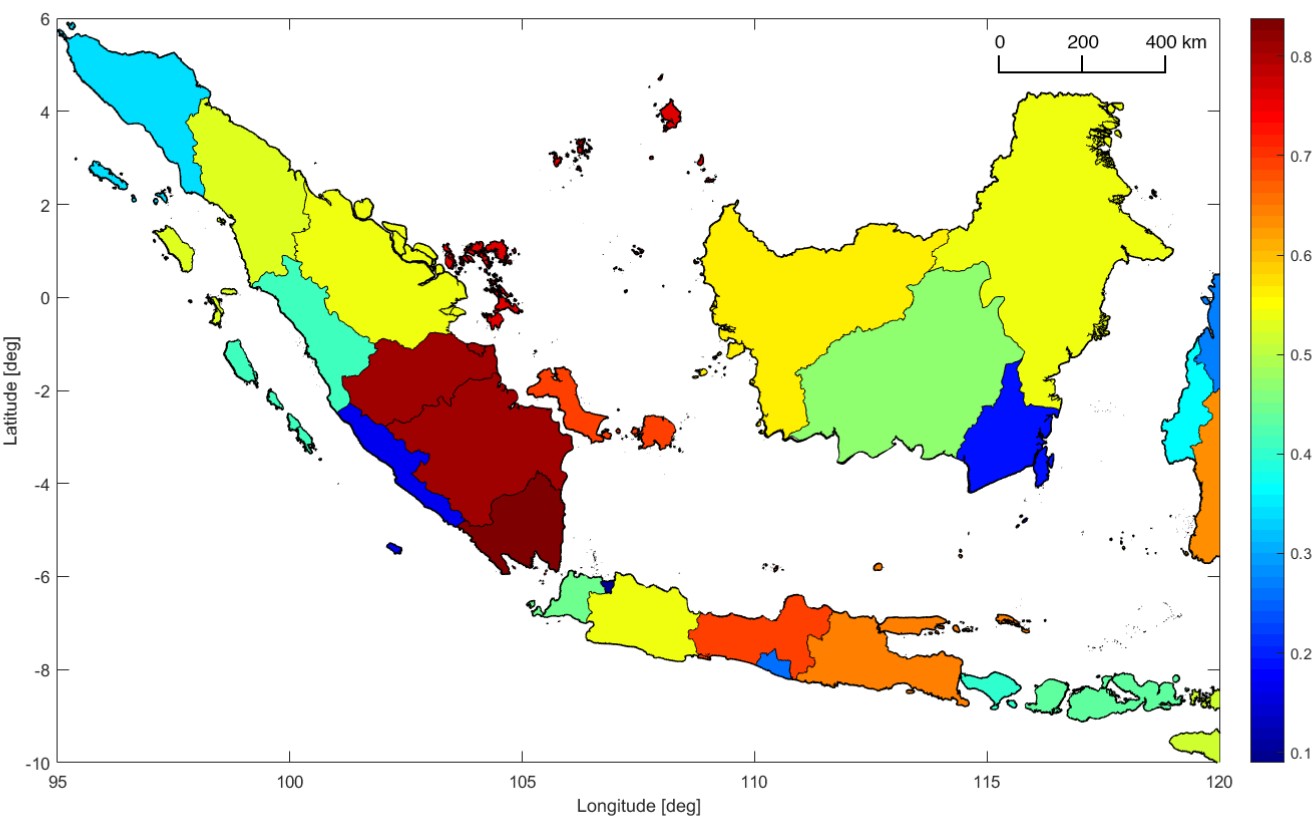

**Figure 7.** Coefficient of variation (CV) of the loss rate inside provinces in western Indonesia. Color denotes the CV.

that it depends directly on properties of the portfolio and a precalculated hazard variability (see Section 3), but does not require on-the-fly integral presampling, such as the one used by certain general purpose adaptive variance reduction schemes (Press and Farrar, 1990; Jadach, 2003).

## 4.1 Risk Location Index Mapping Table

5 We store an array containing all unique geographical locations on which ground motion is simulated, and another array storing the sampled location indices per risk item. Table 2 illustrates the concept. Each column of the table corresponds to a location set representing a valid realization of location uncertainty for the entire portfolio. To combine unequal sample sizes for risk items without introducing bias due to overemphasis of a subset of a sample, we restrict the sample size to powers of two. The full sample can then be repeated in the mapping table.

**Table 2.** Risk Location Index Mapping Table. Rows correspond to individual risk items, showing sampled grid point indices. Each column represents a possible spatial distribution of the portfolio. Risk item 1 has the maximum location sample size of $n_{\max} = 4$, but risk items 2 and 3 only have a sample size of 2 and 1, respectively.

| Risk Item Index | Sample Size | Sample 1 | Sample 2 | Sample 3 | Sample 4 |
|---|---|---|---|---|---|
| 1 | 4 | 43 | 13 | 31 | 51 |
| 2 | 2 | 23 | 18 | 23 | 18 |
| 3 | 1 | 51 | 51 | 51 | 51 |

## 4.2 Criterion I: Coefficient of Variation of Loss Rate

The first criterion is based on the CV of loss rate within a zone (see Section 3), hereafter denoted by $\mathrm{CV_z}$. The values of $\mathrm{CV_z}$ can be precomputed for all administrative geographical zones, and therefore the evaluation of this criterion can be implemented in a very efficient manner.

The number of samples $n_\mathrm{L}$ due to criterion I is defined piecewise:

$$n_\mathrm{L}^* = \begin{cases} 1, & \text{if } \mathrm{CV_z} \leq t_\mathrm{l}, \\ \frac{n_{\max}-1}{t_\mathrm{u}-t_\mathrm{l}} \cdot \mathrm{CV_z} + 1, & \text{if } \mathrm{CV_z} \in (t_\mathrm{l}, t_\mathrm{u}), \\ n_{\max}, & \text{if } \mathrm{CV_z} \geq t_\mathrm{u}. \end{cases} \tag{13}$$

Here, $t_\mathrm{l}$ and $t_\mathrm{u}$ are the lower and upper threshold values. $n_{\max}$ represents the maximum used sample size. We round $n_\mathrm{L}^*$ up to the next higher power of two to obtain the final $n_\mathrm{L}$. The criterion is shown in Figure 8 for the example $t_\mathrm{l} = 0.1$, $t_\mathrm{u} = 0.4$ and $n_{\max} = 16$. In our final implementation, $t_\mathrm{l}$ and $t_\mathrm{u}$ are chosen adaptively as empirical quantiles of the CV distribution ($\mathrm{CV}_{0.4}$

for $t_\mathrm{l}$ and $\mathrm{CV}_{0.6}$ for $t_\mathrm{u}$, i.e. the 40% and 60% percentiles) of the loss rate of all administrative zones of a model (see Section 3), which was found to be a reasonable choice for our test cases with the aid of an extensive parameter study (see Section 5.1).

## 4.3 Criterion II: Number of Risk Items

The second criterion involves two steps. The first step defines a maximum sample size for the entire portfolio depending on the total number of risk items $n_\mathrm{r}$ in the portfolio and a threshold $t_\mathrm{p}$ as

$$n_\mathrm{R}^\dagger = \begin{cases} -\frac{n_{\max}-1}{\log(t_\mathrm{p}-1)} \cdot \log(n_\mathrm{r}-1) + n_{\max}, & \text{if } n_\mathrm{r} < t_\mathrm{p}, \\ 1, & \text{if } n_\mathrm{r} \geq t_\mathrm{p}, \end{cases} \tag{14}$$

which is then used to obtain a maximum sample size per zone, depending on the number of risk items in a zone $n_\mathrm{z}$ and a threshold $t_\mathrm{z}$:

$$n_\mathrm{R}^* = \begin{cases} -\frac{n_\mathrm{R}^\dagger-1}{t_\mathrm{z}-1} \cdot (n_\mathrm{z}-1) + n_\mathrm{R}^\dagger, & \text{if } n_\mathrm{z} < t_\mathrm{z}, \\ 1, & \text{if } n_\mathrm{z} \geq t_\mathrm{z}. \end{cases} \tag{15}$$

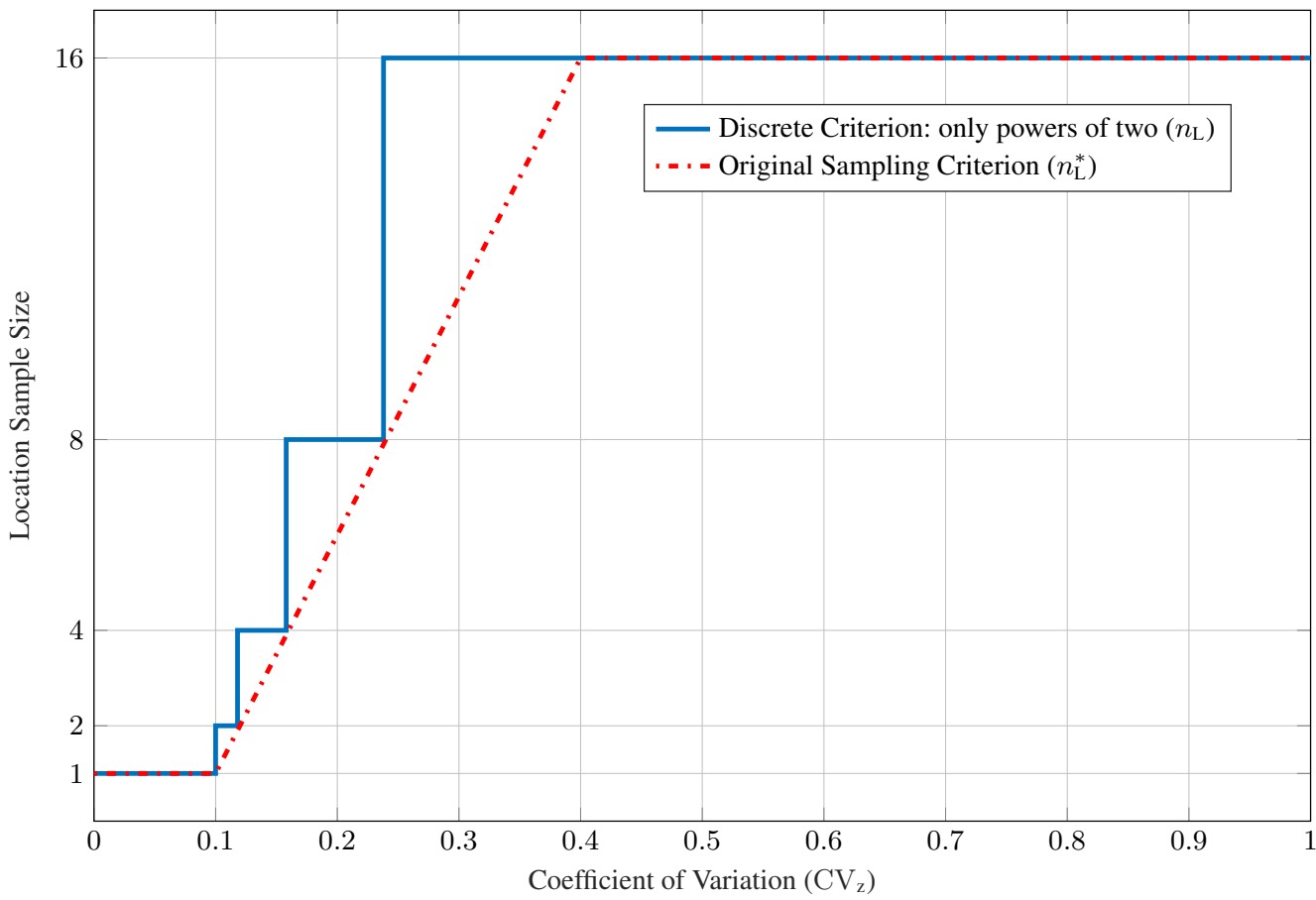

**Figure 8.** Criterion I: Number of samples per zone depending on coefficient of variation. The discrete realization of the criterion (limited to powers of two) is shown in blue, while the red line represents the theoretical linear behavior.

We round $n_R^*$ up to the next higher power of two to obtain the final $n_R$. Figures 9 and 10 illustrate this criterion for $t_p = 10000$, $t_z = 100$ and $n_{max} = 16$. In this study, $t_p$ is chosen to be 10000 and $t_z$ is set adaptively to equal the number of grid points of the weighted location uncertainty sampling grid (see Section 2.2) inside each administrative zone. The design of this criterion is based on the results of a previous study, in which we systematically investigated the effect of location uncertainty and loss aggregation due to spatial clustering of risk items for a large range of different portfolios. It was found that location uncertainty typically has a neglectable effect for very large portfolios and a roughly flat value distribution (Scheingraber and Käser, 2019).

### 4.4 Criterion III: Value Distribution

The third criterion depends on the relative insured values of risk items ("sum insured", SI). Risk items are sorted with respect to their SI, and the index of their sorted order $I_r$ is used along with a threshold index $t_i$ to determine the maximum sample size

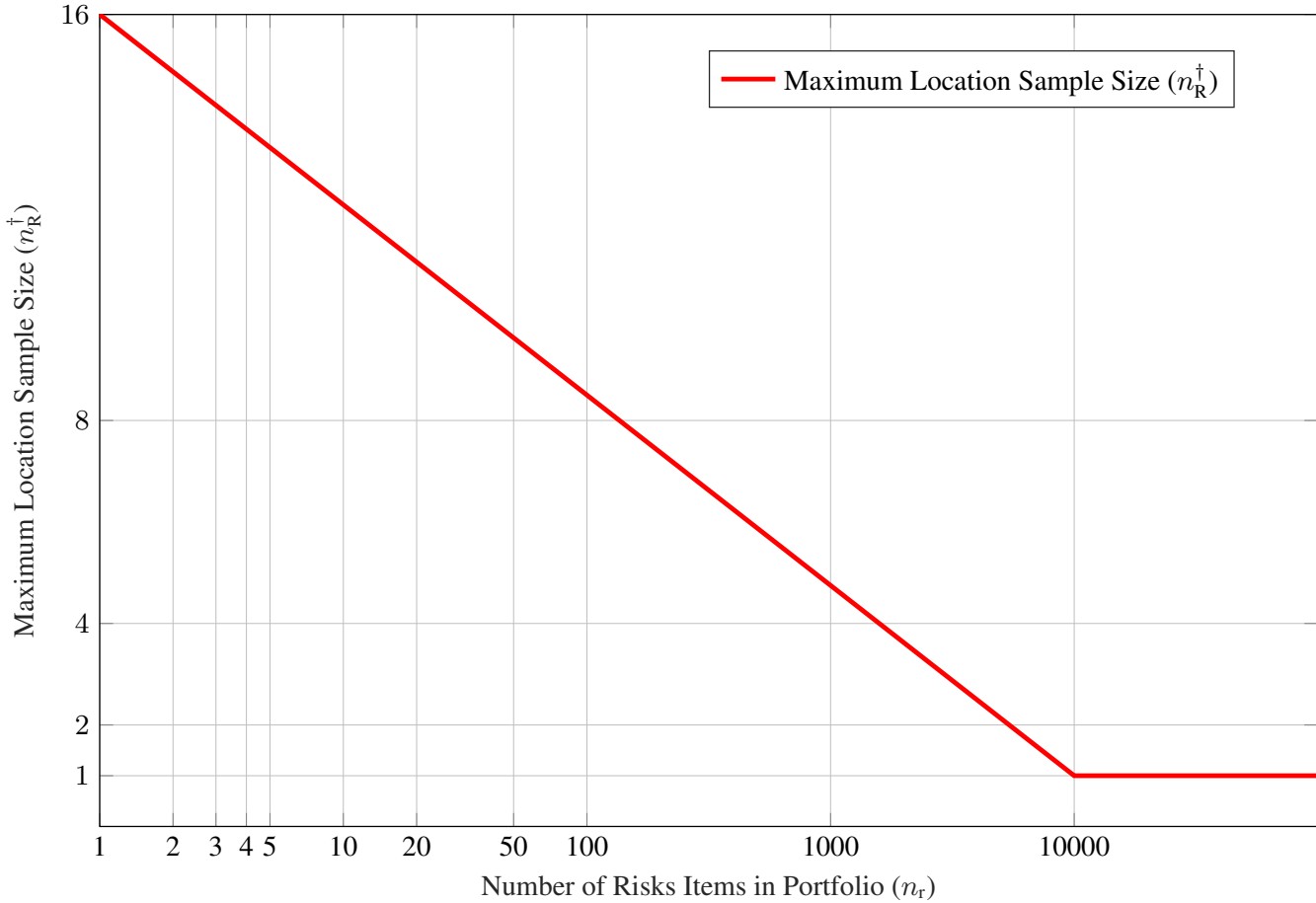

**Figure 9.** Criterion II.a: Number of samples per zone depending on the number of risk items in the portfolio. Note that we do not round up to the next higher power of two, since this plot illustrates Equation 14, which is an intermediate step.

per risk item:

$$
n_V^* = \begin{cases} -\frac{n_{max}-1}{t_i-1} \cdot (I_r - 1) + n_{max}, & \text{if } I_r < t_i, \\ 1, & \text{if } I_r \geq t_i. \end{cases} \tag{16}
$$

We round $n_V^*$ up to the next higher power of two to obtain the final $n_V$. Figure 11 illustrates this criterion for $t_i = 6$ and $n_{max} = 16$. In this study, for $t_i$ we adaptively set the index of the first risk item which has a SI higher than the mean of all risk items.

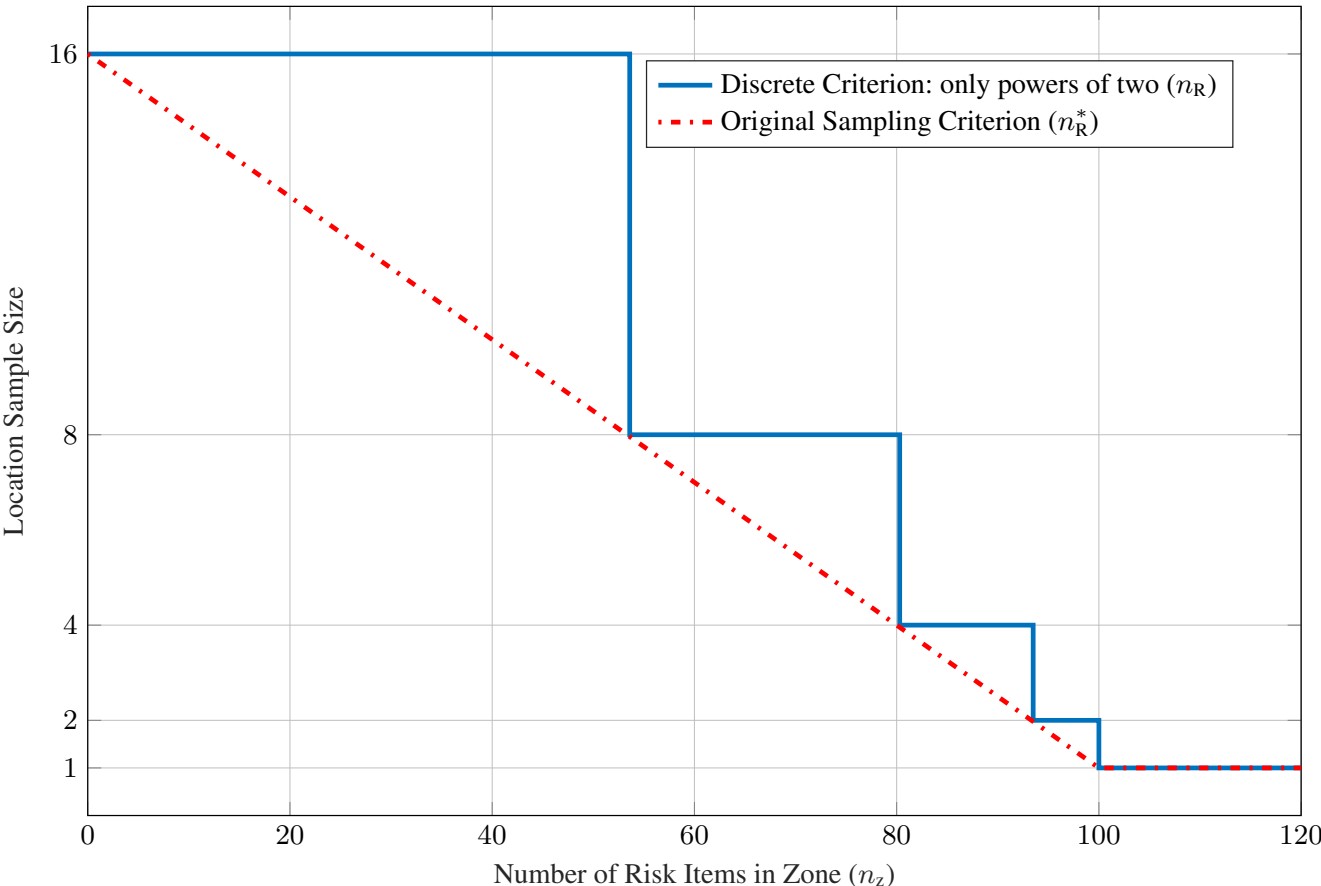

**Figure 10.** Criterion II.b: Maximum number of samples per zone depending on the number of risk items in an administrative zone. The discrete realization of the criterion (limited to powers of two) is shown in blue, while the red line represents the theoretical linear behavior.

## 4.5 Combination of Criteria

The final sample size for a specific risk item is then given by the minimum of the three criteria:

$$n = \min\{n_{\mathrm{L}}, n_{\mathrm{R}}, n_{\mathrm{V}}\}. \tag{17}$$

The rationale behind this decision is that any of the criteria can separately predict that a particular risk item has a low impact on loss uncertainty. For example, if a risk item with an unknown coordinate has a low insured value, it has a relatively low impact on loss uncertainty, even if the variation of hazard or loss rate within the corresponding administrative zone is high, and thus a small location uncertainty sample size can be used. Vice versa, the impact of location uncertainty is limited if a risk item with an unknown coordinate has a high insured value but the hazard within the corresponding administrative zone is relatively flat.

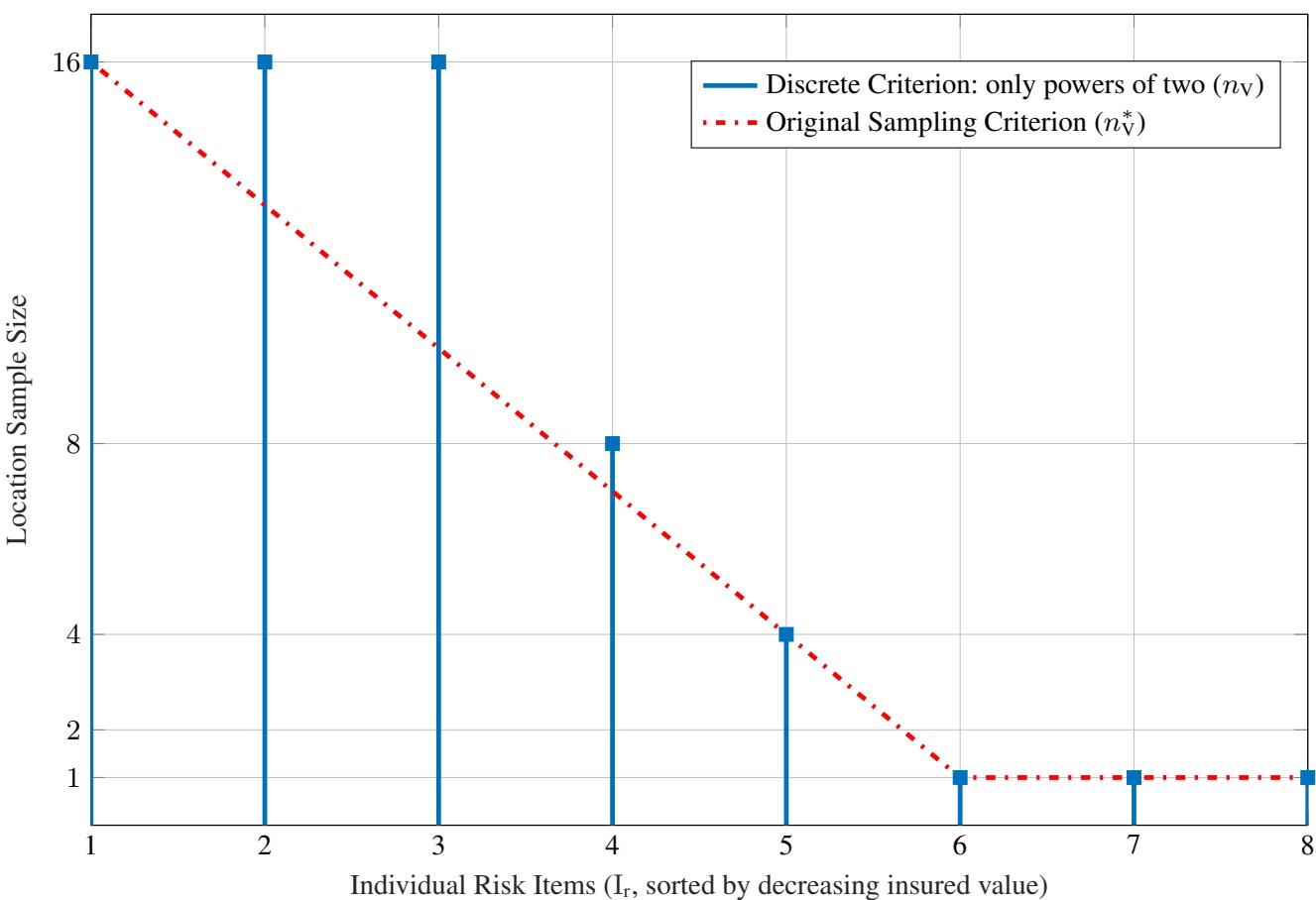

**Figure 11.** Criterion III: Number of samples per zone depending on the insured values of the risk items. The discrete realization of the criterion (limited to powers of two) is shown in blue, while the red line represents the theoretical linear behavior.

Furthermore, loss uncertainty is also limited if a portfolio contains a very high number of total risk items or the number of risk items belonging to an administrative zone is high compared to the number of grid points within this zone.

## 5 Results

In this section, the variance reduction and speedup obtained with the proposed adaptive location uncertainty sampling scheme is analyzed using the western Indonesia hazard model described in Section 3.1 in conjunction with a vulnerability model for regional building stock composition. To this end, loss frequency curves are computed for the synthetic portfolios described in Section 2.4 with simple MC as well as the adaptive scheme. The convergence and relative standard errors are evaluated against

the number of unique hazard locations used for the loss calculation by either approach and the associated required runtime is compared.

## 5.1 Spatial Variation Parameter Study

We first analyze the performance of the adaptive sampling scheme for different values of the lower ($t_l$) and upper ($t_u$) threshold

parameters for the spatial variation of loss rate in an administrative zone in comparison to simple sampling. In simple MC, all risk items get the same location uncertainty sample size $n_{max}$ and there is no restriction to powers of two. For this parameter, we use values of $n_{max} = 32, 64, 96, 128, 160, 192, 224, 256$ in order to obtain a smooth curve with a high number of support points.

For the adaptive variance reduction scheme, the sample size is restricted to powers of two and is determined for each risk

item individually - potentially smaller than the maximum allowed location uncertainty sample size $n_{max}$ (see Section 4 and Table 2). Since the sample size varies between risk items, for a meaningful comparison with simple MC it is necessary to use a measure of the total effort spent for the treatment of location uncertainty of all risk items. We use the total number of unique hazard locations ($n_{hazard}$) and the runtime spent for the computation of hazard ($t_{hazard}$). While for simple MC all risk items get the maximum sample size $n_{max}$, the adaptive location sampling scheme reduces the sample size for risk items for which

location uncertainty likely has a smaller influence. This means that the adaptive location sampling scheme results in a smaller $n_{hazard}$ than simple MC for the same portfolio and $n_{max}$. Therefore, in order to obtain comparable values for $n_{hazard}$, a larger maximum sample size $n_{max}$ has to be employed for the adaptive scheme than for simple MC. Here, we use $n_{max} = 2^i$ with $i = 5, 6, \ldots, 8$.

For each sample size, the spatial variation threshold parameters are varied over the distribution of CV values, picking quantiles in constant steps of 0.2. The lower threshold $t_l$ is varied from $CV_{0.0}$ to $CV_{0.8}$, and the upper threshold $t_u$ from $CV_{0.2}$ to $CV_{1.0}$. For each combination of $t_l$ and $t_u$, $R = 20$ repeated simulations were performed for each sample size to estimate the respective relative standard error $E_{RSE}$.

In general, for our test cases the scheme works well around $t_l \in [CV_{0.2}; CV_{0.4}]$ in combination with $t_u \in [CV_{0.6}; CV_{0.8}]$.

For example, for a portfolio of 20 risk items and 100% unknown coordinates, Figure 12 shows a logarithmic plot of the relative standard error $E_{RSE}$ of PML at a return period of 100 years against the number of used hazard locations $n_{hazard}$ for some combinations of $t_l$ and $t_u$. The error curves for all combinations of $t_l$ and $t_u$ have the same slope as the curve for simple MC and thus the same convergence order of $\mathcal{O}(n^{-0.5})$. For certain combinations, the error curve is below the curve for simple MC, meaning that in these cases the scheme successfully reduces the variance of the estimation and therefore the associated

standard error.

For the final implementation, we used $t_l = CV_{0.4}$ and $t_u = CV_{0.6}$, which performed best in this parametric study.

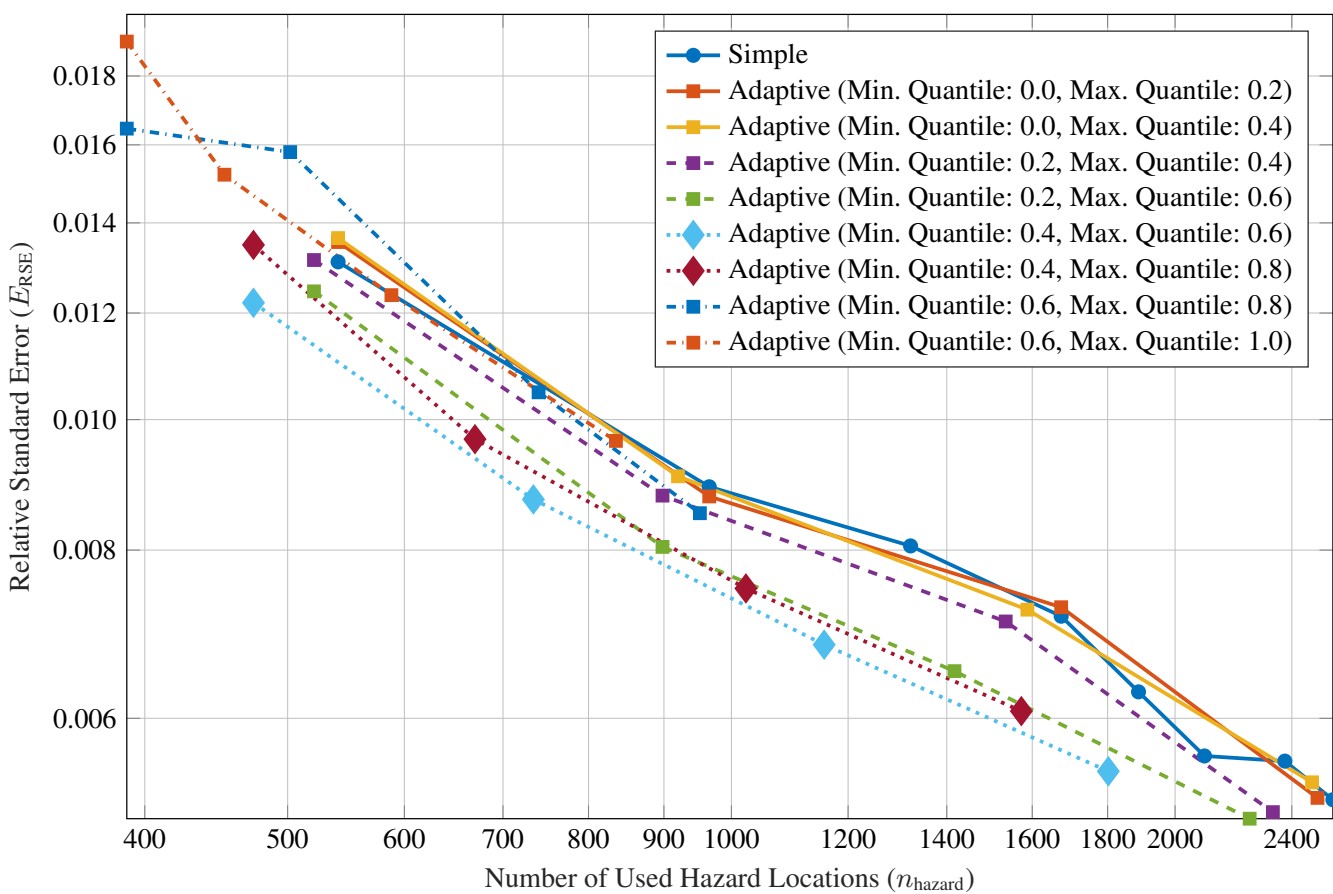

**Figure 12.** Results of a systematic parameter study with the goal of finding good values for the lower threshold ($t_l$) and upper threshold ($t_u$) parameters of Criterion I of the adaptive location uncertainty sampling scheme (see Section 4), based on the distribution of the Coefficient of Variation (CV) of loss rate in administrative zones. This shows a logarithmic plot of relative standard error ($E_{RSE}$) of Probable Maximum Loss (PML) at a return period of 100 years against the number of used hazard locations ($n_{hazard}$) for a portfolio of 20 risk items with 100% unknown coordinates. Color indicates different combinations for the threshold parameters $t_l$ and $t_u$. Quantiles of the CV distribution around $t_l \in [\mathrm{CV}_{0.2}; \mathrm{CV}_{0.4}]$ in combination with $t_u \in [\mathrm{CV}_{0.6}; \mathrm{CV}_{0.8}]$ work best.

## 5.2 Performance of the Final Implementation

We now evaluate the performance of the final implementation of the adaptive scheme, checking if it results in any unwanted systematic bias and investigating variance reduction and speedup for the calculation of PML for different portfolios.

### 5.2.1 Convergence and Bias

Figures 13 and 14 show convergence plots of PML at 100 years return period against the number of used hazard locations $n_{\text{hazard}}$ for portfolios with $n_{\text{r}} = 10$ and $n_{\text{r}} = 100$ risk items, respectively. The left plots depict the results for portfolios with 60% unknown coordinates, the right plots the results for portfolios with 100% unknown coordinates. Simple sampling is shown in blue, the adaptive scheme in red. For all portfolios, the sample size $n$ was varied as $n = 2^i$ with $i = 3, 4, \ldots, 9$. For each sample size and both sampling schemes, $R = 20$ repeated simulations are shown as semi-transparent circles, with solid lines highlighting one individual repetition.

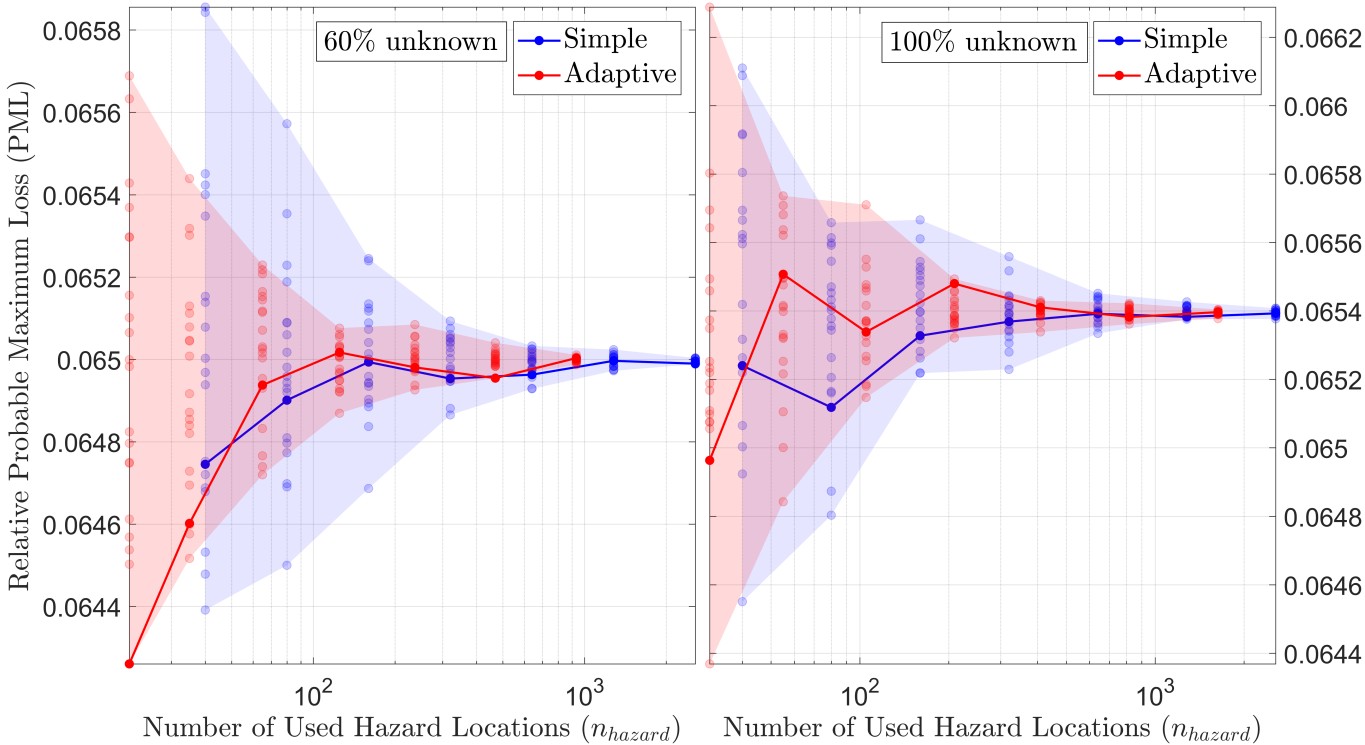

**Figure 13.** Convergence plots showing relative Probable Maximum Loss (PML) at a return period of 100 years against the number of used hazard locations $n_{\text{hazard}}$ for portfolios of $n_{\text{r}} = 10$ risk items with 60% (left plot) and 100% (right plot) unknown coordinates using simple MC (shown in blue) as well as the adaptive scheme (shown in red). Semi-transparent circles depict $R = 20$ repeated simulations for each sample size, solid lines highlight one repetition. The transparently shaded background shows the entire range for each sampling scheme. The plots show that the adaptive scheme scatters less and converges faster to the same result as simple sampling.

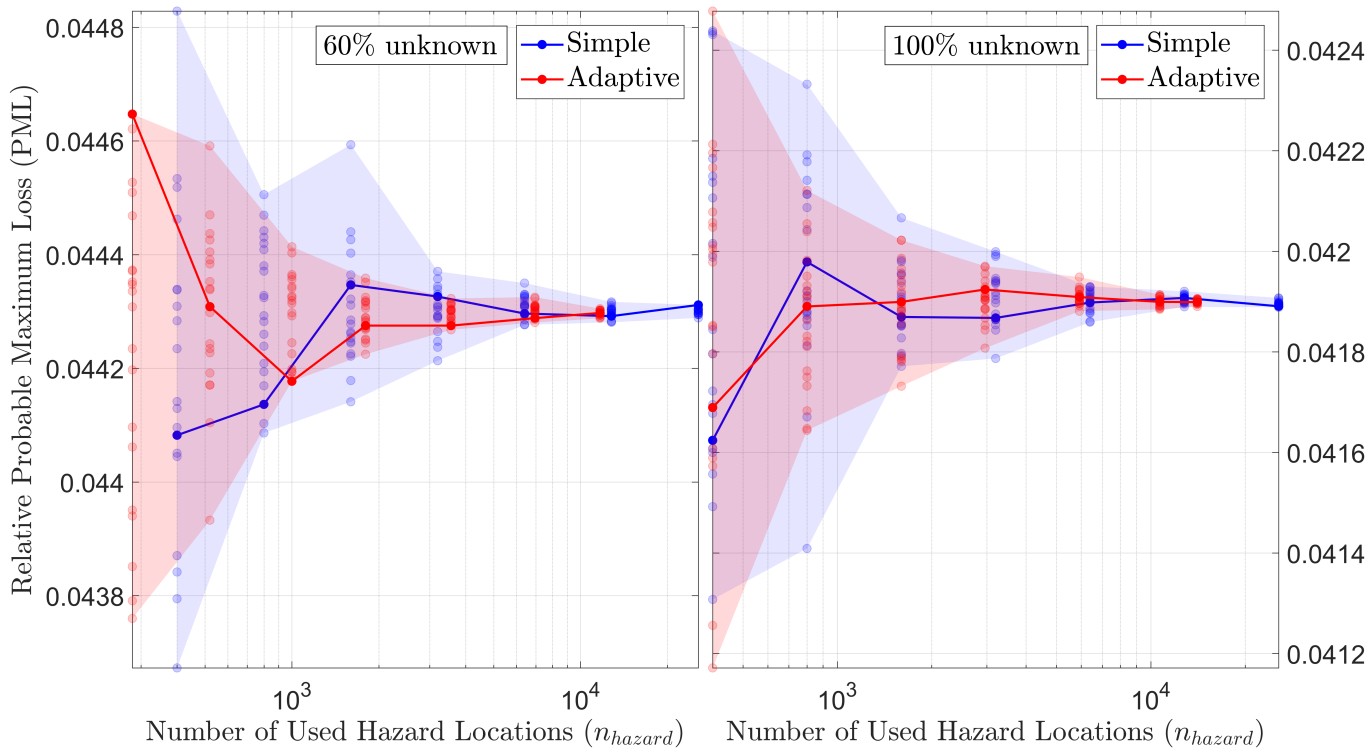

**Figure 14.** Convergence plots showing relative Probable Maximum Loss (PML) at a return period of 100 years against the number of used hazard locations $n_{\text{hazard}}$ for portfolios of $n_{\text{r}} = 100$ risk items with 60% (left plot) and 100% (right plot) unknown coordinates using simple MC (shown in blue) as well as the adaptive scheme (shown in red). Semi-transparent circles depict $R = 20$ repeated simulations for each sample size, solid lines highlight one repetition. The transparently shaded background shows the entire range for each sampling scheme. The plots show that the adaptive scheme scatters less and converges faster to the same result as simple sampling.

The results show that empirically the adaptive scheme converges to the same result as simple MC for our test cases, meaning that the scheme does not result in any systematic bias. It is also apparent that for a given number of used hazard locations $n_{\text{hazard}}$, the relative PML values obtained with the adaptive scheme scatter less than those estimated with simple MC.

### 5.2.2 Variance Reduction and Speedup

For the same portfolios, as analyzed in the previous section, Figure 15 shows logarithmic plots of the relative standard error $E_{\text{RSE}}$ obtained from $R = 20$ repeated simulations against the number of used hazard locations $n_{\text{hazard}}$. Vertical bars depict upper 95% confidence intervals estimated using bootstrapping with 1000 resamples. Simple MC is again shown in blue, the variance reduction sampling scheme in red. While the observed error convergence order of the adaptive scheme remains the same as for simple MC (i.e. $\mathcal{O}(n^{-0.5})$, compare Section 5.1), the error curves are below those for simple MC for all portfolios.

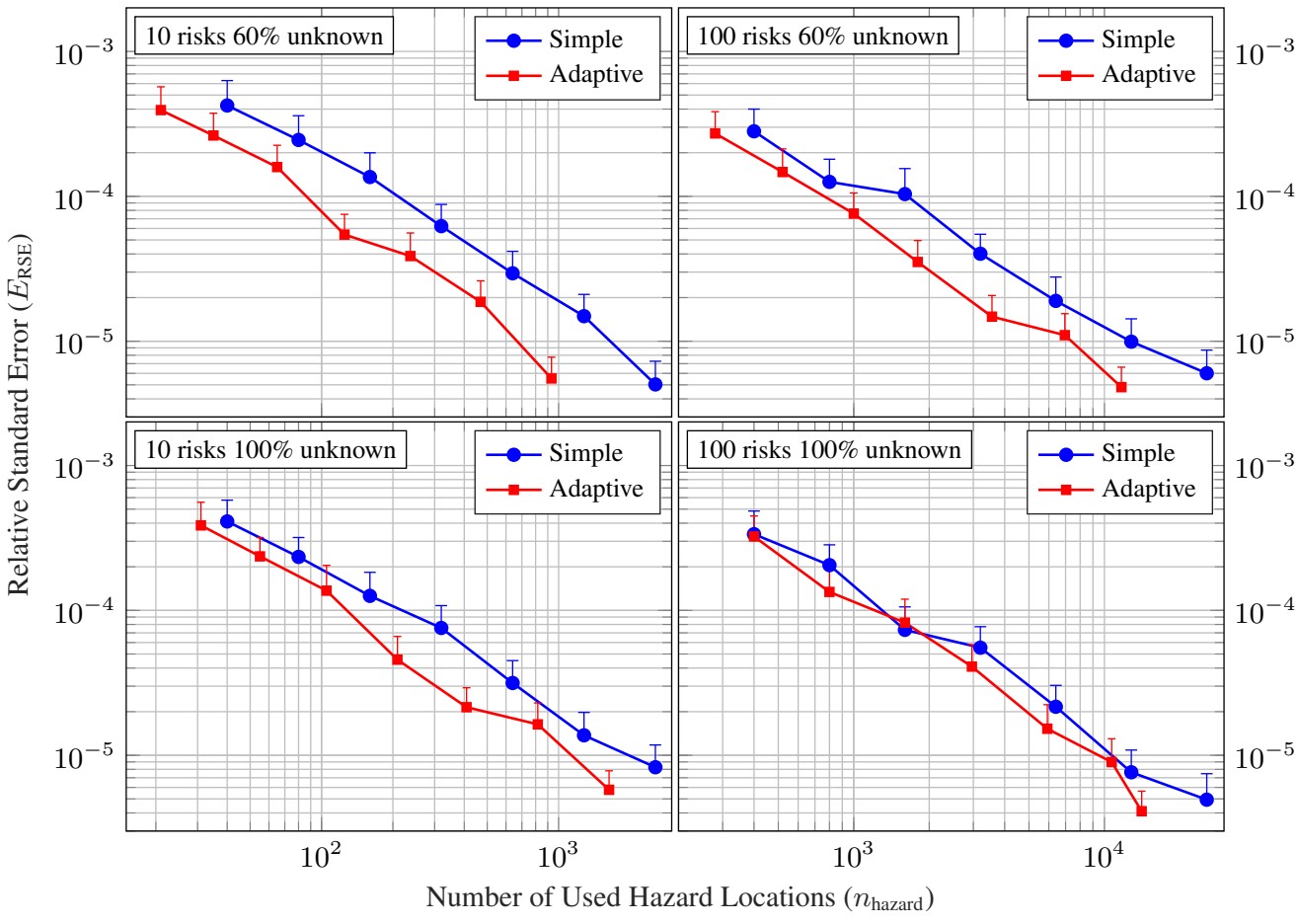

**Figure 15.** Logarithmic plot of relative standard errors $E_{\mathrm{RSE}}$ of Probable Maximum Loss (PML) at a return period of 100 years against the total number of used hazard locations $n_{\mathrm{hazard}}$ for different portfolios with $n_{\mathrm{r}} = 10$ (left plots) and $n_{\mathrm{r}} = 100$ (right plots) risk items and 60% (upper plots) and 100% (lower plots) unknown coordinates. Simple MC is shown in blue, the adaptive variance reduction scheme in red. All $E_{\mathrm{RSE}}$ have been obtained from $R = 20$ repeated simulations, vertical error bars depict upper 95% confidence intervals estimated using bootstrapping with 1000 resamples.

**Table 3.** Mean runtime speedup and standard errors ($S \pm E_{SE}$) of the hazard computation achieved by the adaptive location uncertainty sampling scheme in comparison to simple sampling to obtain relative standard error levels of $\varepsilon_{RSE} = 10^{-4}$ and $\varepsilon_{RSE} = 10^{-5}$, estimated from $R = 20$ repeated simulations. Depending on the portfolio and $\varepsilon_{RSE}$, the mean speedup ranges from 6% to 37%.

| | Speedup (S) | |
| --- | --- | --- |
| Portfolio | $\varepsilon_{RSE} = 10^{-4}$ | $\varepsilon_{RSE} = 10^{-5}$ |
| 10 risk items, 60% unknown coordinates | $1.24 \pm 0.09$ | $1.14 \pm 0.04$ |
| 10 risk items, 100% unknown coordinates | $1.35 \pm 0.06$ | $1.37 \pm 0.09$ |
| 100 risk items, 60% unknown coordinates | $1.08 \pm 0.04$ | $1.06 \pm 0.03$ |
| 100 risk items, 100% unknown coordinates | $1.09 \pm 0.03$ | $1.08 \pm 0.02$ |

The variance reduction quotient (VR, the ratio of the variances of the estimations obtained using simple MC and the adaptive scheme, see Equation 7) varies between portfolios with different number of risk items and fractions of unknown coordinates, but generally increases with growing $n_{hazard}$. For example, for the portfolio with 10 risk items and 60% unknown coordinates, VR is about 6.2 at $n_{hazard} = 10^2$ and increases to 13.2 at $n_{hazard} = 10^3$. For the portfolio with 10 risk items and 100% unknown coordinates, VR $\approx 1.8$ at $n_{hazard} = 10^2$ and 2.2 at $n_{hazard} = 10^3$. For the portfolios with 100 risk items, the situation is similar. For 60% unknown coordinates, VR $\approx 2.4$ at $n_{hazard} = 10^3$ and 3.7 at $n_{hazard} = 10^4$. For 100% unknown coordinates, VR $\approx 1.7$ at $n_{hazard} = 10^3$ and 3.0 at $n_{hazard} = 10^4$.

The obtained variance reduction partially leads to a speedup of the computational runtime to reach a specific relative standard error level $\varepsilon_{RSE}$. Table 3 shows the speedup S of the scheme to reach relative standard error levels of $\varepsilon_{RSE} = 10^{-4}$ and $\varepsilon_{RSE} = 10^{-5}$ for the same portfolios. Depending on the portfolio, the scheme achieves a speedup between 8% and 35% to reach $\varepsilon_{RSE} = 10^{-4}$ and between 6% and 37% to reach $\varepsilon_{RSE} = 10^{-5}$. Note that we obtained these speedup values using a highly optimized seismic hazard and risk analysis framework. We suspect that the scheme can result in a significantly higher speedup for less optimized code, especially if the hazard simulation is not vectorized but contains a loop over locations.

## 6 Conclusions

Seismic risk assessment is associated with a large range of deep uncertainties. For example, the exact location of risks is often unknown due to geocoding issues of address information. In order to provide a holistic view of risk and to be able to communicate the effect of uncertainty effectively to decision makers, all model uncertainties need to be treated. Therefore, in this paper we propose a novel adaptive sampling strategy to efficiently treat this location uncertainty using a seismic hazard and risk model for western Indonesia. The adaptive scheme considers three criteria to decide how often an unknown risk coordinate has to be sampled within a known administrative zone: (1) the loss rate variation within the zone, (2) the number of risks within the zone and (3) the individual value of the risk. As the variation of hazard can vary quite strong not only between different

administrative geographical zones, but also between different return periods, we use the spatial variation of loss rate which displays a similar pattern as the variation of hazard, but is independent of the return period. Furthermore, the total number of risks in the corresponding administrative zone, as well as the value (importance) of the risk with respect to the entire portfolio are considered by the adaptive scheme.

We investigated the performance of the scheme for a large range of sample sizes using different synthetic portfolios of different levels of unknown risk locations. We have found that the scheme successfully reduces the expected error, i.e. it reaches the same error levels as simple Monte Carlo with less samples of potential risk locations. This results in lower memory requirements and a moderate but appreciable runtime speedup to reach a desired level of reliability when computing loss frequency curves - a critical measure of risk in the insurance industry. The scheme helps to obtain a holistic view of risk including the associated uncertainties and could also be applied to other natural perils, such as probabilistic wind and flood models. This should help to avoid blind trust in probabilistic risk assessment.

While the proposed scheme already successfully reduces the variance of loss frequency curve estimations, future improvements in the treatment of uncertainty in PSRA are conceivable and necessary. The effect of modeling assumptions and the often poor data quality needs to be investigated further. The computation might become yet more efficient by the application of variance reduction techniques to other uncertainties, for example in the ground motion and vulnerability models. Moreover, it would be essential to investigate the relative importance of location uncertainty in comparison to these other uncertainty types.

*Code and data availability.* The hazard model used for this work is based on the publicly available United States Geological Survey (USGS) South-East Asia seismic hazard model by Petersen et al. (2007), as described in Section 3.1. The seismic risk model and analysis software used for this work is proprietary and not publicly available.

*Author contributions.* CS implemented the location uncertainty sampling schema, analyzed the seismic hazard variation and sampling schema performance, and wrote the paper. MK conceptualized the sampling methodology, provided guidance for the analysis and for writing the paper, and revised the paper.

*Competing interests.* This work has been supported by a scholarship provided by Munich Re to the first author, who has worked on the project in the course of his doctoral studies at Ludwig Maximilian University of Munich. The second author works as senior seismic risk modeler at Munich Re and is a professor at Ludwig Maximilian University of Munich. Munich Re sells reinsurance products and services based on PSHA and PSRA and therefore has a high interest in modeling the inherent uncertainties within these methods, but did not influence the results of this study in any way.

*Acknowledgements.* This work was supported by a scholarship provided to the first author by Munich Re. We thank the reviewers for their insightful comments, which helped to considerably improve the original manuscript.

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
