# Peer review of "Spatial Seismic Hazard Variation and Adaptive Sampling of Portfolio Location Uncertainty in Probabilistic Seismic Risk Analysis"

_Natural Hazards and Earth System Sciences, 2019_

## Referee Comment (RC1) · Gerasimos Papadopoulos (Referee) · 15 Oct 2019

This is a quite interesting paper dealing with the Spatial Seismic Hazard Variation and Adaptive Sampling of Portfolio Location Uncertainty in Probabilistic Seismic Risk Analysis. The paper is original, well-written and organized. It deserves publication after minor improvement in the Hazard Model. A few specific comments on this chapter follow. p. 6 Hazard Model There is need to provide more details on the seismogenic sources introduced in the model. p. 6 Write "United States Geological Survey" instead of "United States Geological Service". p. 6. "Site conditions are based on topographic

slope (Wald and Allen, 2007)". But site conditions are also dependent on soil conditions. How do you consider this? p. 6. "The CV is the lowest in Kalimantan (< 0.1) due to the absence of any known or modeled crustal faults". It is hard to follow this statement. Therefore, once more there is need to provide more details on the seismogenic sources introduced in the model.
* * *

---

## Referee Comment (RC2) · Anonymous Referee #2 · 18 Oct 2019

This is an interesting paper, well written, illustrated and argued which can be published as is.

---

## Short Comment (SC1) · 20 Oct 2019

In this paper the authors analyze spatial seismic hazard and loss rate variation inside administrative geographical xones in western Indonesia. They propose a novel sampling scheme to improve the efficiency of stochastic portfolio location uncertainty treatment. This paper is well written and treats an interesting subject. However, the paper has a highly technical content dealing with insurance issues and can not be easily read by people who don't have experience in this subject. I would have appreciated more details about the seismic hazard assessment and PGA CV.

---

## Referee Comment (RC3) · Robert J. Geller (Referee) · 8 Nov 2019

Review of "Spatial Seismic Hazard Variation and Adaptive Sampling of Portfolio Location Uncertainty in Probabilistic Seismic Risk Analysis" by Christoph Scheingraber and Martin Käser Referee: Robert J. Geller (anonymity waived)

This manuscript (ms) presents a method for stochastic treatment of portfolio location uncertainty in Probabilistic Seismic Risk Analysis (PSRA). PSRA rests on Probabilistic Seismic Hazard Analysis (PSHA). So the evaluation of this ms is a two-step process.

[Figure]

First, the validity of PSHA must be verified. If this can be done, then the merits of this particular ms must be evaluated.

It might seem odd that a referee is saying in 2019 that the validity of PSHA must be verified, as it is a widely used method and probably a majority of scientists and risk assessors support it. But the mere existence of a consensus means nothing in science. Objective and quantitative testing must be conducted to show validity. However, since the first paper by Cornell (1968), PSHA has become a ubiquitously used method without ever having been objectively vetted and verified.

Recently Mulargia et al. (2017) suggested that the use of PSHA should be abandoned. They pointed out that the basic assumption of PSHA, namely that the frequency of occurrence of past earthquakes can be conflated with the probability of future earthquakes, has never been validated and is probably wrong (also see Stein et al., 2012; Kagan et al., 2012).

Notwithstanding the fact that PSHA has been shown to be fundamentally flawed, many workers, including the authors of this ms, continue to use PSHA. However, this is in no way a justification for accepting such mss for publication. Science is based on data, not on consensus. I therefore recommend that all such mss, including the present ms by Scheingraber and Käser, should be rejected, unless and until they can refute the criticisms of Mulargia et al (2017).

The above recommendation may seem harsh to these authors, as innumerable papers on PSHA or based on PSHA continue to be routinely published. But unless at some point papers on PSHA start to be rejected, they will go on being published forever.

Specific comments on this paper:

1) This referee's impression is that PSHA gives ever more unreliable results as the grid is made finer and finer. Notwithstanding the inherent flaws in PSHA, if the regions were made larger and larger the central limit theorem would probably mean the results were

more safely usable. Maybe this is like the tradeoff between stability and resolution in inverse theory.

2) The quality of the writing is generally good, but there is one typo on page 25. The family name of the first author is "Petersen" not "Mark Petersen." Ditto for the other coauthors. And in the body of the text this paper needs to be called out as "Petersen et al.," not "Mark Petersen et al."

3) Both authors list "Munich Re" as an affiliation on the title page, and the acknowledgements list "Munich Re" as a sponsor. Thus (see line 20 on page 22) the declaration of "no competing interests" is incorrect and must be rewritten. Munich Re sells consulting services and insurance products based on PSRA, and this fact must be appropriately stated in the "competing interests" declaration. The biomedical field is far ahead of the physical sciences in this regard. See the following links. I suggest the authors consult an expert in research ethics to make sure their declaration is in conformity with currents standards. http://www.cell.com/pb/assets/raw/shared/forms/di_form.pdf https://www.nature.com/authors/editorial_policies/competing.html

References Cornell, C.A., 1968. Engineering seismic risk analysis, Bull. Seismol. Soc. Am., 58, 1583-1606.

Kagan, Y.Y., Jackson, D.D., Geller, R.J., 2012. Characteristic earthquake model, 1884-2011, RIP. Seismological Research Letters 83, 951-953.

Mulargia, F., Stark, P.B., Geller, R.J., 2017. Why is probabilistic seismic hazard analysis (PSHA) still used?, Physics of the Earth and Planetary Interiors 264, 63–75.

Stein, S., Geller, R.J., Liu, M., 2012. Why earthquake hazard maps often fail and what to do about it. Tectonophysics 562-563, 1-25.

---

## Author Comment (AC1) · 20 Dec 2019

Thank you for your helpful review. Please find our answers to each of your comments below:

» "This is a quite interesting paper dealing with the Spatial Seismic Hazard Variation and Adaptive Sampling of Portfolio Location Uncertainty in Probabilistic Seismic Risk Analysis. The paper is original, well-written and organized."

Thank you.

» "p. 6 Hazard Model There is need to provide more details on the seismogenic sources introduced in the model."

We understand that the description of the hazard model was too brief, even though it is not the main topic of our manuscript. In the now updated manuscript, we have therefore improved and extended the corresponding Section 3.1 ("Hazard Model"). We have provided information on the results of the original USGS hazard model, and have described the differences to the original USGS model in more detail. For instance, we have added paragraphs on the modeling of the subduction zone geometry and events on the subduction zone as well as on the ground motion model logic tree. Furthermore, we now show and discuss a hazard map produced using our model.

» "p. 6 Write "United States Geological Survey" instead of "United States Geological Service"."

We corrected the text to "United States Geological Survey".

» "p. 6. "Site conditions are based on topographic slope (Wald and Allen, 2007)". But site conditions are also dependent on soil conditions. How do you consider this?"

We acknowledge that the initial description of our site conditions (or soil classes) was too brief. While they are based on the approach proposed by Wald and Allen (2007), they have also been refined locally to consider areas of soft soil such as river beds. In the updated manuscript, we now describe this.

» "p. 6. "The CV is the lowest in Kalimantan (< 0.1) due to the absence of any known or modeled crustal faults". It is hard to follow this statement. Therefore, once more there is need to provide more details on the seismogenic sources introduced in the model."

We see that it was hard to follow the statement about Kalimantan and have therefore – in addition to elaborating more on the hazard model – extended the corresponding paragraph in Section 3.1. We now explain that only homogenous gridded seismicity is used in this area, and also explain why this is the case.

---

## Author Comment (AC2) · 20 Dec 2019

Thank you for your review and positive feedback.
* * *

---

## Author Comment (AC3) · 20 Dec 2019

Thank you for this comment.

We understand that it can be hard to follow the technical details of a highly specialized paper such as this one, in which due to the format we can only provide a concise description of the general methodology of seismic hazard and risk assessment. While we believe that it would clearly be out of scope to explain all details of this scientific area, in our updated manuscript we provide some textbook suggestion for further study

for the interested reader at the end of Section 2.1:

Page 3, Section 2.1 Probabilistic Seismic Hazard and Risk Analysis, Lines 18: "For more details on PSHA and PSRA, we refer to the comprehensive textbook of McGuire (2004)."

In addition, Section 3.1 has been almost entirely rewritten to elaborate more on the utilized hazard model.

---

## Author Comment (AC4) · 20 Dec 2019

Thank you for this review and for sharing your view of PSHA. Please find our answer to each of your comments below.

Comment:

» [. . .] I therefore recommend that all such mss, including the present ms by Scheingraber and Käser, should be rejected, unless and until they can refute the criticisms of Mulargia et al (2017). The above recommendation may seem harsh to these authors,

as innumerable papers on PSHA or based on PSHA continue to be routinely published. But unless at some point papers on PSHA start to be rejected, they will go on being published forever.

Answer:

This comment does not refer to the actual content of our paper, but to the method of PSHA in general. The referenced critique is concerned about the theoretical or physical "validity of PSHA", but we are more interested in the practical benefits the method provides to society.

PSHA – since its first publication by Cornell in 1968 – has been improved in many aspects, especially with regard to its inherent uncertainties. Its benefits are broadly recognized by scientists and practitioners working in PSHA and related fields such as seismic risk analysis and seismic engineering. No better approach has been proposed, which is probably why PSHA continues to be used. To quote the famous statistician George Box: "All models are wrong, but some are useful".

Having co-authored all of the cited papers, the reviewer Robert Geller is a vehement critic of PSHA. The paper co-authored with Kagan and Jackson from 2012 – and cited to support this critique of PSHA – does in fact not even mention PSHA. It criticizes theories in seismology like "earthquake cycle", "seismic gap", and "characteristic earthquake". We agree on most statements of this opinion paper, but do not see any direct or tangible criticism of the general framework of modern event-based PSHA.

Concerning the two other cited papers (from 2012 and 2017, also co-authored by this reviewer), we again agree with their main points that PSHA cannot provide an accurate measure of hazard, and that it must not be used with blind trust (Quote: "Geoscience consultants feed information into the PSHA machine, the crank is turned, and a seemingly objective number pops out."). Since we developed our "PSHA machine" ourselves, we are well aware of its weaknesses and dependence on assumptions that sometimes are barely supported by reliable data collected over time scales long enough for certain seismological needs. One should keep in mind PSHA has its justification exactly in that it can be used to analyze the influence of purely constraint assumptions and quantify the ranges of possible hazard results. The output of PSHA is inherently based on probabilities and therefore it is always possible to find individual examples – as in the suggested papers – that do not nicely fit the model. In any case, abandoning the method entirely and relying only on qualitative judgement does not seem to be a wise recommendation; it is neither practical nor legally viable, because e.g. engineering and financial decisions and regulations require quantitative input.

However, we agree to the argument that the flaws and limits of PSHA and the uncertainties of its results have to be communicated effectively in order to provide a holistic view of risk and avoid "blind trust" in PSHA outputs. This is precisely why our work is concerned with the computationally efficient treatment of uncertainties in PSHA and PSRA, allowing to analyze the impact of poorly constraint or unknown parameters on risk assessment and to come to an informed decision. In this paper in particular, we analyze one of these aspects of uncertainty related to seismic risk assessment: insured portfolios often have poor geospatial data quality. We show how the huge uncertainty space can be investigated efficiently by using variance reduction methods, i.e. adaptive sampling strategies instead of simple Monte Carlo simulation. PSHA is just used to get an input parameter for the sampling method we propose for seismic risk analysis, and our method can help to validate PSHA and PSRA more effectively.

We are therefore a bit surprised that Robert Geller did not recognize that our work is actually in line with most of his argument. We assume that our holistic view of PSHA and PSRA was not made clear enough in the abstract and early sections of our original manuscript. To improve this, in our updated manuscript we have added some clear statements about the weaknesses and limits of PSHA and PSRA:

- Page 1, Abstract, Lines 2-4: "The available seismological data is often limited, resulting in many uncertainties and assumptions. The situation is further aggravated by the sometimes poor data quality with regard to insured portfolios."

- Page 1, Abstract, Lines 16-17: "The results show that the scheme can improve the efficiency of the estimation of loss frequency curves, and may thereby help to spread the treatment and communication of uncertainty in Probabilistic Seismic Risk Analysis."

- Page 1, Section 1 Introduction, Lines 19-20: "Seismic risk analysis is widely used in academia and industry to model the possible consequences of future earthquake events, but is often associated with poor data quality, resulting in the necessity for many assumptions and a wide range of deep uncertainties (Goda and Ren, 2010). The treatment and communication of uncertainties is highly important for informed decision making and a holistic view of risk (Tesfamariam et al., 2010; Cox, 2012; Bier and Lin, 2013)."

- Page 2, Section 2.1 Probabilistic Seismic Hazard and Risk Analysis, Lines 27-28: "PSRA is based on Probabilistic Seismic Hazard Analysis (PSHA; Cornell, 1968; Senior Seismic Hazard Committee, 1997), which relies on a number of assumptions outlined in the following."

- Page 23, Section 6 Conclusions, Lines 2-4: "Seismic risk assessment is associated with a large range of deep uncertainties. For example, the exact location of risks is often unknown due to geocoding issues of address information. In order to provide a holistic view of risk and to be able to communicate the effect of uncertainty effectively to decision makers, all model uncertainties need to be treated. Therefore, in this paper we propose [. . .]"

- Page 25, Section 6 Conclusions, Lines 14-15: "The effect of modeling assumptions and the often poor data quality needs to be investigated further."

Comment:

» 1) This referee's impression is that PSHA gives ever more unreliable results as the grid is made finer and finer. Notwithstanding the inherent flaws in PSHA, if the regions were made larger and larger the central limit theorem would probably mean the results

were more safely usable. Maybe this is like the tradeoff between stability and resolution in inverse theory.

Answer:

Unfortunately, it is unclear to us to which "grid" this comment refers to. In our paper we are mainly referring to "grid" in the sense of a weighted irregular grid that represents potential locations of insured assets. This concept is explained in our paper and is necessary to spatially distribute insured values given only on administrative geographical zones. This grid is not related to the spatial accuracy of analyzing the seismic ground motion or seismic hazard output. It is also not related to another grid which is typically used to model potential earthquake epicenters within an area source using the method of gridded seismicity. Therefore, we believe this comment comes from a misunderstanding of the term "grid" as it is used in the context of this paper.

Comment:

» 2) The quality of the writing is generally good, but there is one typo on page 25. The family name of the first author is "Petersen" not "Mark Petersen." Ditto for the other coauthors. And in the body of the text this paper needs to be called out as "Petersen et al.," not "Mark Petersen et al."

Answer:

Thanks for pointing this out, we have corrected the typo in the citation of Petersen et al. (2007).

Comment:

» Both authors list "Munich Re" as an affiliation on the title page, and the acknowledgements list "Munich Re" as a sponsor. Thus (see line 20 on page 22) the declaration of "no competing interests" is incorrect and must be rewritten. Munich Re sells consulting services and insurance products based on PSRA, and this fact must be appropriately stated in the "competing interests" declaration. [. . .]

Answer:

We think that this comment is misleading, as one might conclude that Munich Re potentially profits in any material sense from having provided financial support for this academic research. There was no business-related influence on this paper and there is no financial interest of Munich Re in publishing scientific papers. In fact, Munich Re is highly interested in making academia aware of some of the crucial problems arising in Probabilistic Seismic Risk Assessment (PSRA), as well as in trying to solve them in a way of best scientific practice.

While working on this project, the first author has not been affiliated with Munich Re, but has been a PhD student at Ludwig Maximilian University of Munich and funded by Munich Re. It is the right and duty of a PhD-student to publish scientific results and discuss the applied approaches and findings with the scientific community. Since the first author has not been affiliated with Munich Re while working on this project, we have removed this affiliation and apologize for the confusion this might have caused. To clarify this and the role of Munich Re in this research, we adjusted the "competing interests" declaration accordingly.

---

## Referee Report (RR1)

[referee-annotated manuscript omitted]

---

## Author Response (AR2)

**Author´s Response to the Editor (Point-by-Point Reply to the Comments)**

**nhess-2019-110**    Submitted on 03 Apr 2019
**Spatial Seismic Hazard Variation and Adaptive Sampling of Portfolio Location Uncertainty in Probabilistic Seismic Risk Analysis**
Christoph Scheingraber and Martin Käser

Manuscript Type: Research article
Handling Editor: Filippos Vallianatos, fvallian@geol.uoa.gr

Editor Decision: Publish subject to minor revisions (review by editor) (09 May 2020) by Filippos Vallianatos

Comments to the Author:
Be so kind to introduce into the text the suggestion given by reviewer.

Non-public comments to the Author:
Your revised version will be reviewed only by the editor.

**Response to **Report #1**

Submitted on 08 Apr 2020
Anonymous Referee #1

We are thankful for this review and positive feedback.

**Response to **Report #2**

Submitted on 09 May 2020
Anonymous Referee #4

>> *The manuscript deals with Probabilistic Seismic Risk Analysis, applied in Indonesia, also implementing Probabilistic Seismic Hazard Assessment. The subject is of wide interest, both for Academics and for Insurance Companies, and the work is worth being published. It is a well-written manuscript. […] Thus, the manuscript can be considered for publication in Natural Hazards and Earth System Sciences after minor revisions.*

We are thankful for this positive and constructive review. We have carefully gone through the comments and the annotated manuscript provided by this reviewer and incorporated all suggestions and corrections. Please find our detailed answers to each of the comments below:

>> *However, general expressions, such as (among others) "seismological data is often limited" or "poor data quality" must be avoided and specified.*

We have improved these general expressions in order to more precisely explain in which respect seismological data is limited for the purpose of seismic risk assessment (see our marked-up revised manuscript; also see our response to the detailed comments below).

*>> In addition, other methods of Probabilistic Seismic Hazard Assessment must be mentioned and the selection of the applied method should be justified.*

In the revised manuscript, we have added an overview of other methods of seismic hazard assessment and justified the selection made for this study (see Section 2.1 of our marked-up revised manuscript; also see our responses to the related detailed comments #2 and #3 below).

*>> Detailed comments, corrections and additions are included, mainly as sticky motes, in the pdf file: nhess-2019-110-manuscript-version2_reviewer1.pdf. Some of the main comments (also included in the .pdf file) are:*

*>> 1. In all maps add a scale in kilometers (km)*

In the revised manuscript, we have added a scale in kilometers to all maps.

*>> 2. Page 1, Line 2: "seismological data is often limited". Which seismological data are often limited? Please be more precise. Seismological networks are densified, so this sentence might be misleading.*

See answer to comment #3.

*>> 3. Page 1, Lines 2-3: "resulting in many uncertainties and assumptions". This is a general statement which is misleading. You refer in general in "seismological data". But, for example, source parameters are quite all determined nowadays. You have to clarify to what uncertainties and assumptions you refer to. Try to avoid general and misleading comments.*

We agree that the sentence referred to by comments #2 and #3 might have been misleading and have therefore improved it: "The available ground motion data - especially for strong and infrequent earthquakes - are often limited to a few decades, resulting in incomplete earthquake catalogues and related uncertainties and assumptions. The situation is further aggravated by the sometimes poor data quality with regard to insured portfolios."

*>> 4. Page 1, Lines 7-8: ", but also between different return periods for a fixed zone." This is self-evident and is always the case in regions with high seismicity. Delete this part of the sentence.*

We have deleted this part of the sentence as requested.

*>> 5. Page 1, Line 20: "poor data quality". Again, which data have poor quality? I urge the authors to avoid such general and misleading comments.*

We have improved this statement as following: "[…] is often limited by the availability of reliable earthquake event ground motion data over a longer period of time, resulting in the necessity for many assumptions and a wide range of deep uncertainties (Goda and Ren, 2010)."

*>> 6. Page 6, Lines 21-22: "where p is a perturbation factor set to 0:2, which is consistent with the characteristics of many real portfolios". Provide more details for p and add references.*

We have added more details and some references: "[...] where p is a perturbation factor controlling the variation of insured values among individual risk items in the portfolio. For this study, we set p to 0.2. This value corresponds to the characteristics we observe in many real residential portfolios and is in accordance with the assumption of a relatively flat value distribution, which is commonly made when modeling the value distribution of residential building stock (e.g. Kleist et al. 2006; Okada et al. 2011)."

*>> 7. Page 7, Lines 6-7: "For this study, we use a custom seismic risk model based on the South-East Asia hazard model by Petersen et al. (2007) of the United States Geological Survey (USGS)". The authors should first describe different models and methods applied for Probabilistic Seismic Hazard Assessment in several studies. For example they should refer to the Extreme Values method (e.g. Papadopoulou-Vrynioti et al., 2013; Pavlou et al., 2013) and to the SEISRISK III method (e.g. Benders and Perkins, 1987). Following, the authors should explain why (based on which criteria) they chose to use the model by Petersen et al. (2007).*

As proposed, in the revised manuscript we added an overview of different methods of seismic hazard and risk analysis to Section 2.1, and give reason why the approach of event-based stochastic simulation is well-suited for the purposes of the insurance industry.
We have also explained why we have chosen the model of Petersen et al. (2007) to Section 3.1.

*>> 8. Page 8, Line 1: "Sumatra, Java, and Kalimantan (the Indonesian sector on the island of Borneo". Page 8, Last Line "Jakarta". All toponyms must be added to Figure 2*

We have added all toponyms.

*>> 9. Figure 2: Add all toponyms to the map (Sumatra, Java, Kalimantan, Jakarta etc)*

We have added all toponyms.

We have also incorporated all further suggestions this reviewer provided as notes in the annotated manuscript. See our revised, annotated manuscript for details.

[revised manuscript text omitted]